# Significant mass loss in the accumulation area of the Adamello glacier indicated by the chronology of a 46 m ice core

Daniela Festi[1,2], Margit Schwikowski [3,4,5], Valter Maggi [6,7], Klaus Oeggl [2], Theo Manuel Jenk[3,4]

[1]Faculty of Sciences, Free University of Bozen-Bolzano, 37100 Bozen, Italy

[2]Department of Botany, University of Innsbruck, A-6020 Innsbruck, Austria

[3]Laboratory of Environmental Chemistry, Paul Scherrer Institute, CH-5232 Villigen PSI, Switzerland

[4]Oeschger Centre for Climate Change Research, University of Bern, CH-3012 Bern, Switzerland

[5]Department of Chemistry and Biochemistry, University of Bern, CH-3012 Bern, Switzerland

[6]Dipartimento di Scienze della Terra, Università Milano Bicocca, 20126 Milano, Italy

[7]National Institute of Nuclear Physics, Milano-Bicocca section, 20126 Milano, Italy

*Correspondence to:* Daniela Festi (Daniela.Festi.Dr@gmail.com)

**Abstract.** Dating glaciers is an arduous yet essential task in ice core studies, which becomes even more challenging when the glacier is experiencing mass loss in the accumulation zone as result of climate warming, leading to an older ice surface of unknown age. In this context, we dated a 46 m deep ice core from the Central Italian Alps retrieved in 2016 from the Adamello glacier in the locality Pian di Neve (3100 m a.s.l.). Here we present a timescale for the core obtained by integrating results from the analyses of the radionuclides $^{210}$Pb and $^{137}$Cs with annual layer counting derived from pollen and refractory black carbon concentrations. Our results clearly indicate that the surface of the glacier is older than the drilling date of 2016 by about 20 years and that the 46 m ice core reaches back to around 1944. For the period of 1995-2016 the mass balance at the drilling site (former accumulation zone) decreased on average of about 1 m w.e. a$^{-1}$ compared to the period 1963-1986. Despite the severe mass loss affecting this glacier even in the former accumulation zone, we show that it is possible to obtain a reliable timescale for such a temperate glacier using black carbon and pollen seasonality in combination with radionuclides $^{210}$Pb and $^{137}$Cs. Our results are therefore very encouraging and open new perspectives on the potential of such glaciers as informative palaeoarchives.

**Keywords:** annual layer counting, pollen, black carbon, $^{210}$Pb, temperate glacier, Alps

## 1       Introduction

Ice core studies from mid-latitude mountain glaciers are essential to infer recent climate variability and anthropogenic impact on a regional scale, but they are challenging because these ice masses are, with the exception of very few sites, mostly represented by temperate, or polythermal glaciers, in the better case scenario. Notoriously, the ongoing climate warming is globally causing a progressive reduction in ice bodies (Zemp et al., 2015) and already seriously compromises the climatic and environmental signal embedded in the ice of the most thermally unstable glaciers (Zhang et al., 2015).  Particularly because of the most recent strong warming, such ice masses

may further experience years with negative mass balance even in what had formerly been the accumulation zone. This may cause a surface loss of annual layers, leaving the ice surface to be of unknown age at the time of sampling. The surface age is a major anchoring point for annual layer counting, and without that information dating of an ice core is further complicated, yielding in the best case a floating chronology (Zhang et al., 2015). Because of meltwater percolation occurring in temperate glaciers, proxies such as soluble ions and stable isotopes can be disturbed, making annual layer counting impossible when the seasonality in the signal is lost (Eichler et al., 2001). Thus, the current state of signal preservation in glaciers affected by the warming needs thorough testing, and an urge exists to retrieve these valuable archives of the past before they will be permanently compromised, or even completely vanished. To date, relatively few ice cores from temperate high elevation glaciers have successfully been dated (von Gunten et al., 1982; Naftz et al. 1996, Kang et al., 2015; Pavlova et al., 2015; Neff et al., 2012 Kaspari et al., 2020; Gäggeler et al., 2020). Among them, an ice core from Silvretta Glacier (Eastern Alps, Switzerland; Pavlova et al., 2015).

In this frame, we investigated ADA16, a 46 m ice core drilled at the plateau Pian di Neve of the Adamello glacier (3100 m a.s.l., Italian Alps), located around 80 km south-west of the above mentioned Silvretta glacier. Ice temperatures at the nearby Alto dell'Ortles Glacier (3,859 m a.s.l.) indicated temperate ice with around 0°C from surface to a depth of 30 m, while the ice below was still cold with temperatures reaching -3°C close to bedrock at 74 m depth (Gabrielli et al., 2010). With Pian di Neve (3100 m a.s.l.) being located in the same region, affected by similar climatic conditions, but with a far larger ice thickness, a similar trend in ice temperatures - the presence of temperate ice in the upper part and colder ice temperatures below - is not unlikely. While seismic analyses, do confirm the absence of melt water at the base of the glacier (Picotti et al., 2017), temperate ice conditions are however likely to exist to greater depth compared to the Alto dell'Ortles Glacier considering their difference in altitude. First encouraging results from the ADA16 core were reported by Di Stefano et al. (2019), who detected a distinct peak in $^{137}$Cs activity, which was attributed to the maximum fallout from surface nuclear bomb testing in the year 1963 ($32.0 \pm 0.3$ m depth below surface). Another peak in $^{137}$Cs at a depth of 9.5 m was hypothesised to reflect the signal of the 1986 Chernobyl accident.

Here, we present new records of pollen, refractory black carbon and $^{210}$Pb from analyses performed on the ADA16 ice core. The term refractory black carbon (rBC) is used for black carbon (BC) measured by incandescence methods (Petzold et al., 2013). Pollen and rBC were selected because their concentrations were shown to be rather robust parameters in terms of signal preservation even under temperate ice conditions and the potential influence from melt water percolation. With the signal of seasonal variability at least reasonably preserved, they proved to be particularly useful for the counting of annual layers in temperate ice (Kaspari et al., 2020; Takeuchi at al., 2019; Festi et al., 2017; Pavlova et al., 2015; Nakazawa,studio 2004). Also, both Pb and Cs were found to be reasonably well preserved, i.e. not easily relocated or removed by percolating meltwater (Avak et al., 2018; Avak et al., 2019; also see Gäggeler et al. 2020 for $^{210}$Pb specifically). The main goal of our study was to establish a robust chronology for the ADA16 ice core by combining three independent dating methods, i.e. radiometric dating with $^{210}$Pb, annual layer counting based on different proxies, and time markers identified by $^{137}$Cs maxima.

## 2    Methods

### 2.1    Study site and ice core drilling

The Adamello is the largest glacier in Italy with an extension of 16,3 km$^2$ (Smiraglia and Diolaiuti, 2015) and being located at a relative low elevation of 2500-3400 m a.s.l. (Figure 1) it is currently affected by considerable area loss (19% in the period 1983-2003, Maragno et al., 2009) with recent negative mass balance observed even in what can now be considered the former accumulation zone. This implies that the glacier has currently no persistent accumulation zone. Available glacier mass balance data for the period 2011-2018 show a generally negative mass balance in what is considered to be the former accumulation zone, with slightly positive values only observed for a few particularly favourable years (i.e. 0.2-0.3 m w.e. at the Lobbia glacier in 2013 and 2104; data Meteotrentino.it). More generally, since the 1980s a decrease in snowfalls, snow cover depth and duration has been observed in the area, possibly linked to the increase of air temperature (Bocchiola and Diolaiuti, 2010). The part where the bedrock is deepest below the current glacier surface was selected for ice core drilling; this is located in Pian di Neve, a vast central accumulation plateau at 3,100 m a.s.l. There, a maximum ice thickness of 268±5 m was measured by means of geophysical techniques (Picotti et al., 2017). Although there are no direct information about the current ELA (equilibrium-line altitude) it is likely that the coring site is located below the current ELA (Zebre et al 2021).

The 46 m deep ice core ADA16 was drilled with an electromechanical drill from the 11[th]-13[th] of April 2016 in Pian di Neve (Geographic coordinate system WGS84: 10.52 E, 46.15 N). Prior to drilling, a 3.1 m trench was excavated, removing the fresh winter snow. Hence, drilling and accordingly sampling for the results presented in the following, started from this depth defined as the glacier surface. Drilling operations stopped at 46 m of depth due to wet conditions from percolation/inflow of surface melt water causing technical problems for mechanical drilling. Immediately after coring, the ice core was transported frozen to the Eurocold Lab facilities at the University of Milano Bicocca (Italy) were it was further preserved frozen at -30°C. There, ice core processing and cutting was performed in a -25°C cold room before samples were then shipped frozen to the individual laboratories for analysis.

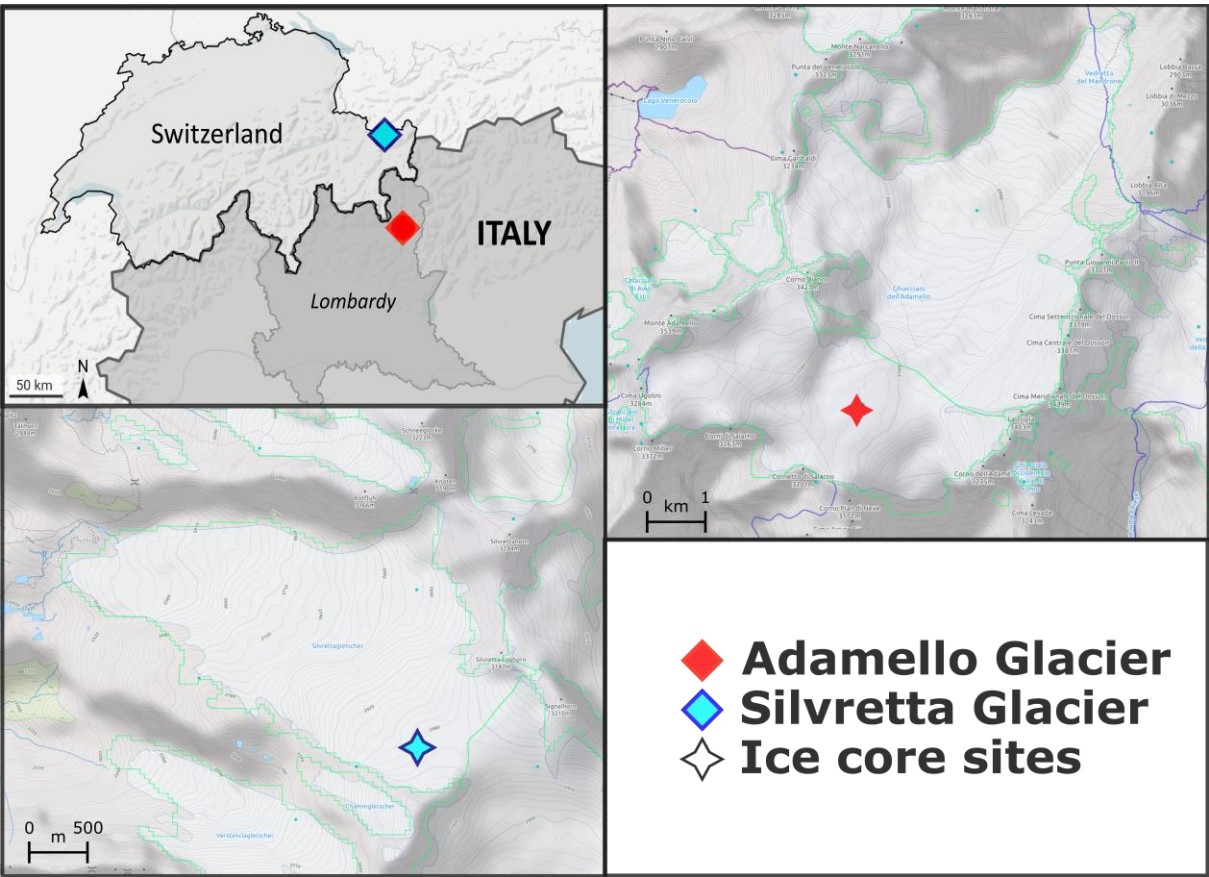

**Figure 1.** Map showing the locations of the Adamello (red diamond) and Silvretta (light-blue diamond) Glaciers) and respective zoom-in maps on ice core drilling sites: Adamello (red star); Silvretta (blue star) . All maps are north-up oriented.

## 2.2 Ice core stratigraphy

The visual stratigraphy was documented in the Eurocold Lab during ice core processing. The density was thereby determined by weighing (precision scale, ±0.01 grams) of precisely cut ice sticks allowing for an exact determination of the volume (2 by 2 cm times length). The density profile indicates an abrupt snow/ice transition at about 4.5 m of depth (i.e. slightly below the starting depth of drilling at 3.1 m), where density shifts from around 0.6 g cm$^{-3}$ to an average of ~0.9 g cm$^{-3}$ (supplementary material, Figure S1). Hence, with a predominant density of 0.9 g cm$^{-3}$, the core is entirely composed of ice, except for the small upper portion of fresh snow/firn. To account for snow/firn densification, depth was converted from meter to meters water equivalent (w.e.) by multiplying with the density obtained from the fit shown in Figure S1.

## 2.3 Pollen and rBC analyses for annual layer counting

The ice core was cut in 536 continuous samples for pollen analyses, with the uppermost 35 m cut at ~10 cm resolution and the deeper 10 m at a ~5 cm resolution. After pollen extraction (Festi et al., 2015, 2019), the complete content of each sample was quantified by manual counting of individually identified pollen and spores (palynomorphs). Pollen and spores identification was performed by means of a light microscope at a magnification of 400x.

For rBC analyses the ADA16 ice core was cut with a ~5 cm resolution in 914 continuous samples which were then placed into pre-cleaned vials and sent frozen to the Paul Scherrer Institute (PSI). The samples were analysed at PSI in September 2018, following the method established by Wendl et al. (2014) and later slightly modified and improved as described in Osmont et al. (2018). In brief, ice samples were melted at room temperature and sonicated in an ultrasonic bath for 25 min before analysis by a Single Particle Soot Photometer (SP2, Droplet Measurement Technologies, USA) (Schwarz et al., 2006; Stephens et al., 2003) coupled with a jet nebulizer (APEX-Q, Elemental Scientific Inc., USA). External calibrations from 0.1 to 50 ppb (linear, $R^2 > 0.999$) were performed daily by preparing fresh dilutions of a BC standard (Aquadag®, Acheson Inc., USA). The liquid flow rate of the APEX-Q was monitored several times per day to avoid changes in the nebulizing efficiency. The instrumental blank was checked between every sample and kept below detection limit (<1 ppb rBC) by rinsing the setup with ultrapure water. Automated sample analysis was performed using a CETAC ASX-520 auto-sampler (CETAC Technologies, USA), programmed to measure each sample until 10000 rBC particles counts were reached with a limiting condition for the total sample measurement time (1 min < measurement time < 30 min). Between each sample the auto-sampler probe was rinsed with ultrapure water until the rBC signal returned to the baseline value (45 sec), and the sample take-up time prior to data acquisition was set to 1.75 min. A small systematic correction as a function of time was performed to account for the time dependent (elapsed time from sonication until analysis) effect from vial-wall adsorption and particle agglomeration.

## 2.4 $^{210}$Pb analyses

$^{210}$Pb is a naturally occurring radionuclide. It forms in the atmosphere via radioactive decay of radon ($^{222}$Rn), which constantly emanates into the atmosphere from the Earth crust where it is produced by the decay of uranium ($^{238}$U). Often applied for nuclear dating of environmental samples such as lake sediments or peat bogs, $^{210}$Pb, attached to aerosol particles, is deposited on glacier surfaces via scavenging with fresh snow. With a half-life of 22.3 years, it allows dating of ice cores over roughly one century. In the ADA16 ice core, $^{210}$Pb activity was determined continuously throughout the core on 29 samples of increasing resolution from ~2 m in the top 20 m to ~1 m for the lower part of the ice core. Following an established method (Gäggeler et al., 1983; Gäggeler et al., 2020), $^{210}$Pb activity was determined via the α-decay of its grand-daughter nuclide $^{210}$Po. In brief, under reducing conditions, polonium was deposited onto the surface of a silver plate immersed into the melted sample. Therefore, ADA16 samples of 200 mL were acidified with 10 mL $HCl_{conc}$ prior to melting. Then, 100 μL of a $^{209}$Po standard with known activity was added as a tracer for the chemical yield. The solution was heated to 90°C for ~10 h and reducing conditions were established by bubbling $SO_2$ gas through the liquid for 3 min. A silver disk (6 mm diameter) was immersed into the continuously mixed liquid. After deposition, the activity on the disk was measured with an α-spectrometer (Enertec Schlumberger 7164) when placed opposite to a silicon semiconductor detector, able to resolve the well separated α-decay energy lines of the $^{209}$Po standard and $^{210}$Po sample (at 4.9 MeV and 5.3 MeV, respectively). To achieve a detection uncertainty of 1%, α-counting was ended after reaching 10000 counts. Chemical yields of ~70% were determined and the sample processing blank used for blank correction was less than 1 mBq kg$^{-1}$, similar to the system background (detection limit).

## 3    Results

### 3.1    Annual layer counting based on pollen and black carbon concentrations

Both, palynomorphs and rBC concentrations show a marked seasonal signal, with 34 synchronous peaks occurring throughout the ADA16 ice core (Figure 2). The synchronicity of rBC and pollen and spores maxima is striking, and a strong indication for the preservation of the seasonality of the signal (Figure 3).

Palynomorphs maxima have an average concentration of 28 poll/mL, ranging from 2.2 to 120 poll/mL, while minima samples mostly contain no pollen and no spores at all (Figure 2, panel A). Extraction of the sub-seasonal signal was attempted according to Festi et al. (2015), performing a Principle Component Analyses (PCA) using all taxa present in more than ten samples. The PCA results showed that 96% of the variance is included in the first principle component, pointing to the fact that no sub-seasonal signal (i.e. spring, early and late summer) is preserved, and implying that a complete flowering year was condensed in a thin layer contained mostly in one sample. For this reason, we assume that each peak of pollen and spores concentration reflects one flowering year (February-September), while palynomorphs free layers reflect the non-flowering season (October to January). Maxima in pollen and spores's concentration were therefore used to identify individual annual layers. In the pollen record, two layers exceptionally rich in palynomorphs stand out at 2.1 and 12.2 m w.e. depth, potentially representing a signal of multiple years condensed in one single layer caused by years of negative mass balance and enrichment of the pollen and spores at the exposed surface.

The seasonal cycle observed for rBC at high altitudes, with summer maxima and minima during the winter season, is mainly controlled by enhanced thermal convection occurring more frequently in summer (Lavanchy et al., 1999). As for the palynomorphs, maxima in rBC were thus similarly used to identify individual annual layers. In the ADA16 ice core, rBC concentrations vary from of 0.1 to ~100 ppb with three exceptional high peaks of ~180 ppb, ~230 ppb and ~770 ppb at 9.3 m w.e., 27.7 m w.e. and 30.1 m w.e. depth, respectively (Figure 2, panel B).

For the dating by annual layer counting (ALC) the year 1963 identified by the peak in [137]Cs at 27.1 m w.e. depth (32 m; Di Stefano et al., 2019) was used as a tie point, i.e. as the starting point for counting (Figure 2, panel A). Counting peaks of palynomorphs and rBC towards the surface and the bottom, yields 34 layers in total (marked with an asterisk in Figure 2). By assigning multiple years (three) to the layer of exceptionally high pollen and spores concentration at 12.2 m w.e. depth, the year 1986 identified by the second peak in [137]Cs at 6.6 m w.e. (Di Stefano et al., 2019) is perfectly matched. Continuing counting pollen and spores maxima all the way up to the glacier surface, a surface age of the core equal to $1993^{+0}_{-3}$ was derived. The three years of uncertainty thereby account for a potential error when also assigning three years to the second pollen rich layer at 2.1 m w.e. depth. Although both palynomorphs rich layers are comparable in their concentration, some uncertainty is certainly justified also considering the fact that no exceptionally high peak in rBC was observed at this depth. In contrast to pollen production, anthropogenic rBC emissions show a downward trend beginning in the second half of the 20[th] century (Sigl et al., 2018), which might also explain the absence of an outstanding peak at 2.1 m w.e. dept.

In summary, the dating by ALC is very robust and strongly suggests the surface of the glacier to be considerably older than 2016, when the core was drilled. An age estimation for the bottom of the core cannot be inferred by ALC because both, rBC and pollen concentrations were low below 36.3 m w.e. depth, corresponding to the year 1956.

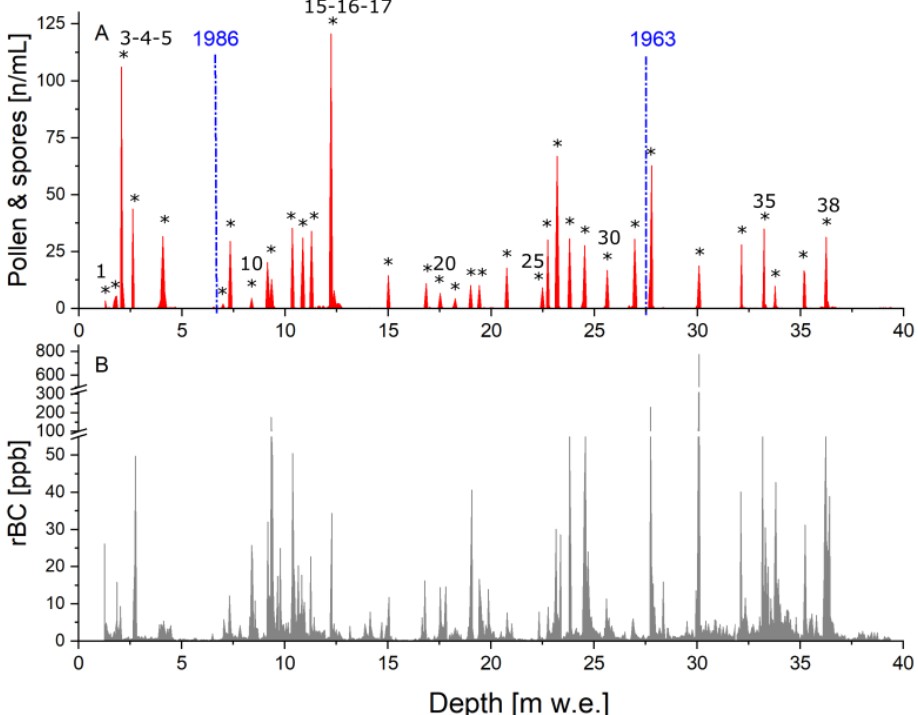

**Figure 2** A) Pollen and spores' (palynomorphs) concentration; B) refractory black carbon (rBC) concentration in the ADA16 core. Blue lines indicate the [137]Cs horizons (Di Stefano and others, 2019), to which the absolute annual layer counting chronology is tied. Asterisks mark every peak of pollen and rBC considered to represent one or more years. Peaks are labelled starting from the top of the core with numbers from 1 to 38.

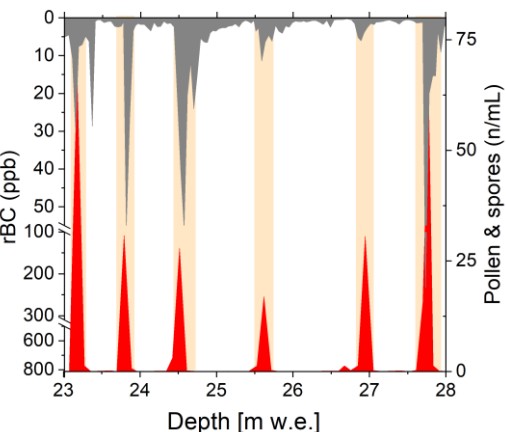

**Figure 3** Zoom in of a 5 m w.e. section of the palynomorphs (pollen and spores) and rBC records presented in Figure 2, highlighting the synchronicity of their peak maxima and minima.

**3.2      [210]Pb record**

The environmental radionuclide $^{210}$Pb proved to be a crucial tool in the dating of temperate glaciers in the past (Kang et al., 2015; Pavlova et al., 2015; Kaspari et al., 2020; Gäggeler et al., 2020). Here, $^{210}$Pb was used as a third, independent dating tool. The Adamello $^{210}$Pb ice core profile did not show a clear exponential decrease in activity concentrations with increasing depth as it is typically observed in glacier ice (supplementary Figure S2). This is usually observed because of the relation between an increase in age with depth and the radioactive decay of $^{210}$Pb; however only in the typical case of a close to constant initial $^{210}$Pb activity concentration and fairly constant annual accumulation rates (Gäggeler et al., 2020). In the Adamello ice core, we further observed an unexpected high value of $692\pm31$ mBq kg$^{-1}$ in the upmost sample, much higher than the mean annual activity of 86 mBq kg$^{-1}$ observed in freshly deposited snow on glaciers in the European Alps (Gäggeler et al., 2020). While the low values of around 10-40 mBq kg$^{-1}$ below the surface sample suggest that $^{210}$Pb has already decayed (T½ = 22.3 a), indicating the ice at this depth to be already several decades old, the high value in the uppermost sample suggests enrichment of the particle-bound $^{210}$Pb in the surface layer. The ADA16 $^{210}$Pb record strongly resembles the $^{210}$Pb profile of the nearby Silvretta (SI) ice core drilled in 2011 (SI, 2930 m asl., Eastern Swiss Alps, Figure 1). In the SI ice core, the 1963 nuclear bomb test horizon was identified by a peak in $^3$H found at 28.9 m w.e. depth (Pavlova et al., 2015). As shown in Figure 5, the SI depth scale was shifted by -1.8 m w.e. to match this time marker observed in both cores. By doing so, a reasonable alignment of the two $^{210}$Pb profiles was achieved, both showing a very similar, characteristic pattern. The SI dating - based on ALC combined with local mass balance data and independently verified by $^{210}$Pb (Pavlova et al., 2015) - was thus directly transferred to the ADA16 ice core. The resulting chronology exactly matches the presumed 1986 $^{137}$Cs peak and is in very close agreement with the ADA16 dating by ALC described in Section 3.1, especially for the period 1963-1986 (within a few years, see Figure 5). This agreement and the resemblance of the ADA16 and SI ice core $^{210}$Pb profiles show that accumulation rates at these two sites of relatively close proximity and similar altitudes are very comparable.

The chronology derived based on $^{210}$Pb suggests that the surface ice at the Adamello drill site was formed in the year $1998 \pm 3$. This agrees with the dating based on ALC (Sect. 3.1), again indicating a significant loss of annual layers at the glacier surface prior to the drilling date in 2016. Likewise, the high activity of 692 mBq kg$^{-1}$ measured in the uppermost sample can only be explained by enrichment in the surface layer. Assuming a mean annual surface activity of 86 mBq kg$^{-1}$ (see previous paragraph) and taking radioactive decay into account, this enrichment corresponds to the activity content of around 13 annual snow/firn layers being lost due to negative mass balance. This represents a lower limit, since $^{210}$Pb removal by drainage is likely, but cannot be quantified.

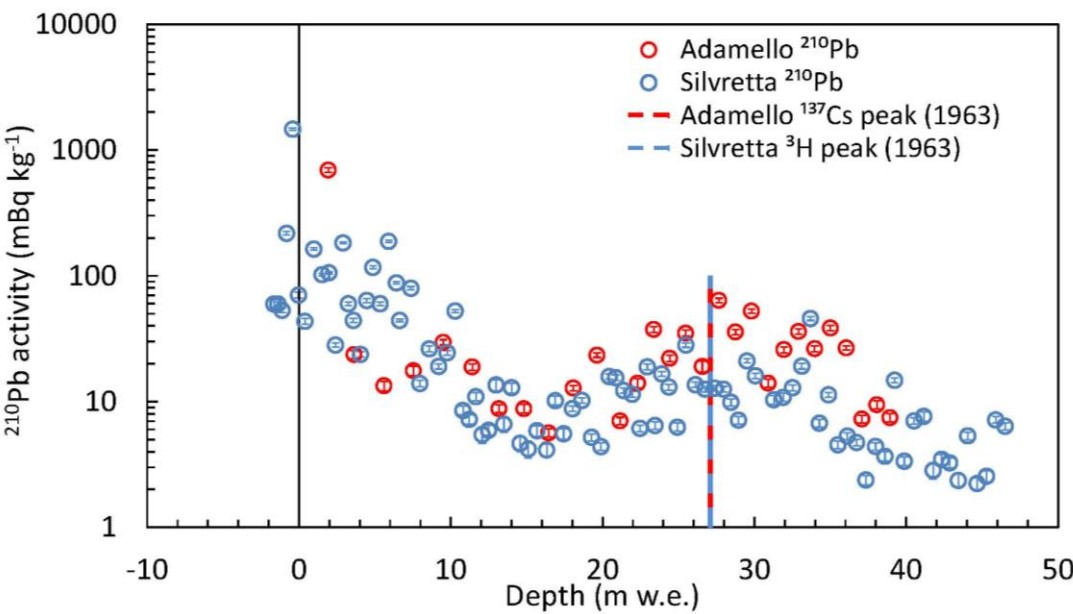

**Figure 4.** [210]Pb profiles of the ADA16-Adamello and Silvretta ice cores, aligned by shifting the Silvretta depth scale by -1.8 m w.e. to match the 1963 horizon observed in both cores.

## 4      Final Adamello chronology

The dating obtained with the three independent methods (ALC, [210]Pb, [137]Cs) is in excellent agreement. The age-depth relationship is particularly reliable in the period between 1963 and 1986, where the precision of the ALC can be evaluated in relation to the anchor points provided by the [137]Cs horizons, and where the [210]Pb timescale, which is tied to the 1963 [137]Cs peak only, freely matches the 1986 [137]Cs peak. Our results confirm previous findings that both palynomorphs (pollen and spores) and BC are not severely influenced by percolating meltwater and that pollen grains tend to accumulate on the ablation surface and are not easily vertically displaced (Pavlova et al., 2015; Festi et al., 2017). We further conclude that even in a glacier heavily affected by summer ablation like the Adamello, a reliable age-depth relationship can still be obtained when working with a combination of dating methods (Figure 5). In the following the main results of this combined approach will be summarized.

**Age of surface.** The two, completely independent methods used to estimate the age of the surface layer delivered an age older than the drilling date of 2016 with $1993^{+0}_{-3}$ (ALC) and $1998\pm3$ ([210]Pb), respectively. This points to the fact that a significant number of annual layers was lost due to negative local mass balance in recent times. Based on the good agreement and our confidence in the dating we can conclude that for about 20 years no net accumulation has been preserved at the drill site. As a matter of fact, the year 1998 also marks the beginning of a period of substantially more negative mass balance on the local Mandrone Glacier (Ranzi et al. 2010; Grossi et al 2012) as well as generally in the Alpine region (Carturan et al., 2013; Huss et al., 2015). Our findings also confirm that the drilling site is likely located below the current ELA, as modelled by Žebre et al (2021) for this sector of the Alps.

**Age of bottom of the core.** Since ALC cannot be applied below 36 m w.e. due to the low rBC and pollen concentrations, our chronology for that part solely relies on the [210]Pb timescale, and reaches a maximum age of 1944±6.

**Annual net accumulation rate for the period 1963-1986.** As already discussed, our timescale is particularly reliable in the 24-year period between 1963 and 1986 corresponding to 20.45 m w.e., allowing to infer an average annual net accumulation rate of 0.85 m w.e. a[-1] (0.9 ± 0.03 m w.e. a[-1] if accounting for layer thinning, see next paragraph). This value is consistent with net accumulation rates at other nearby glaciers, like the Ortles (0.85 m w.e. a[-1] 1963-2011; Gabrielli et al., 2016) and the Silvretta Glacier (0.9 m w.e. a[-1] 1940-2010; Pavlova et al., 2015). Point mass balance reconstructions as derived here for the ADA16 drill site have particular value as they have been shown to reflect changes in climate better than total mass balance or terminus fluctuation (Vincent et al., 2017). Relatedly, it has also been shown that point mass balance changes can reveal clear regional consistencies, which might be of interest for future work in the framework of the comparison with data from the Ortles and Silvretta glaciers.

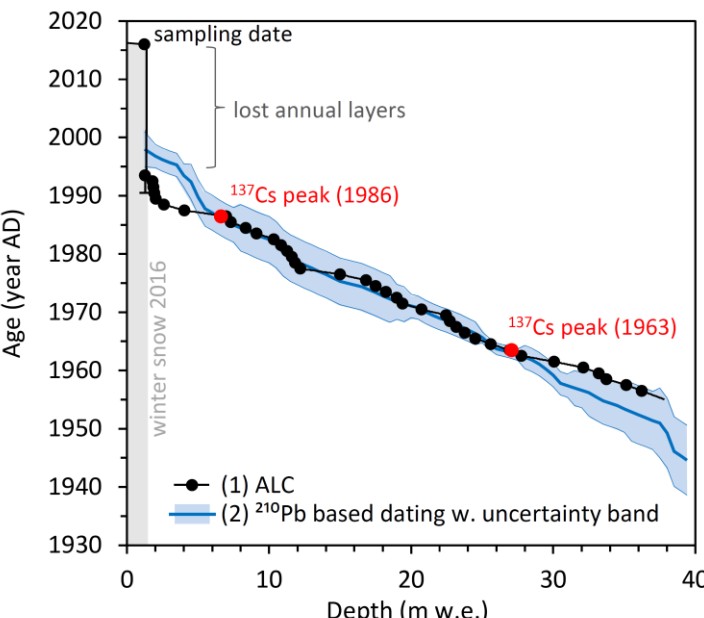

**Figure 5**. Synoptic graphic showing the ADA16 age-depth relationship independently derived from (1) ALC using pollen and rBC concentrations in combination with the distinct time markers from the nuclear accident of Chernobyl (1986) and the nuclear surface bomb testing maximum (1963) identified by peaks in [137]Cs activity and (2) based on [210]Pb (i.e. the age-depth scale with uncertainty of the Silvretta ice core transferred to ADA16, see text).

## 4.1 Modelling the age of Pian di Neve at bedrock

For an estimation of the potential age range accessible by the Adamello ice archive, the one-dimensional Dansgaard-Johnsen ice-flow model was applied (Dansgaard and Johnsen, 1969). For the resulting age-depth relationship estimate shown in Figure 6, model parameters were as follows. Based on the bedrock depth determined by Picotti et al. (2017) using seismic measurements, the value for glacier thickness at the drill site was 265 ± 5 m

(238 ± 4.5 m w.e.). The bottom shear zone thickness was assumed to be 15 % of the glacier thickness. This is slightly lower than the ~20 % typically observed for cold and polythermal high-elevation glaciers (e.g. Jenk et al., 2009; Uglietti et al., 2016; Gabrielli et al., 2016; Licciulli et al., 2020) but likely more reasonable for a temperate glacier (e.g. Kaspari et al., 2020). In any case, because constraining information from dated age horizons is lacking for the bottom part, a relatively large uncertainty of ±10 % was assigned. With these parameter settings, the value

for the annual accumulation rate was found by tuning for a best model-fit to the dated 1986 and 1963 [137]Cs horizons (least squares approach). The dating uncertainty and the uncertainties associated with the pre-set model parameters described above were employed to derive upper and lower bound estimates (to transfer the uncertainty contribution from the uncertainty in ice thickness into age, relative depths were used).

The model - nicely matching the determined bottom age for the ADA16 core and accounting for layer thinning

(vertical strain) – provides us a best estimate of the mean annual accumulation rate at the ADA16 drill site for the period ~1946 to 1986 of 0.9 ± 0.03 m w.e. a$^{-1}$. However, the assumption of steady-state conditions and the complexity of bedrock geometry and glacial flow in the deepest part of high-alpine glaciers strongly limits a realistic modelling of strain rates (and thus age) for the deeper parts, even using the most complex glaciological 3D ice-flow models. In our case, the lack of data for additional constraint in the deeper/older part, the assumption

of steady-state conditions in annual accumulation rates (equal to an average value for the entire period contained in the archive) but only a short time range covered by the 46 m core, causes the derived model-based age-depth relationship to be highly uncertain. Despite that, this estimate is still sufficient to reveal the potential of the site. The Adamello ice archive is very likely to cover the last 1000 years. Being contained in the major part of the total ice thickness (about the upper 240 m of ice; ~220 m w.e.), a millennial-long record should thus be accessible in

high resolution. Also, there is reasonable likelihood for a few thousand more years contained in the remaining ~10 % of ice below.

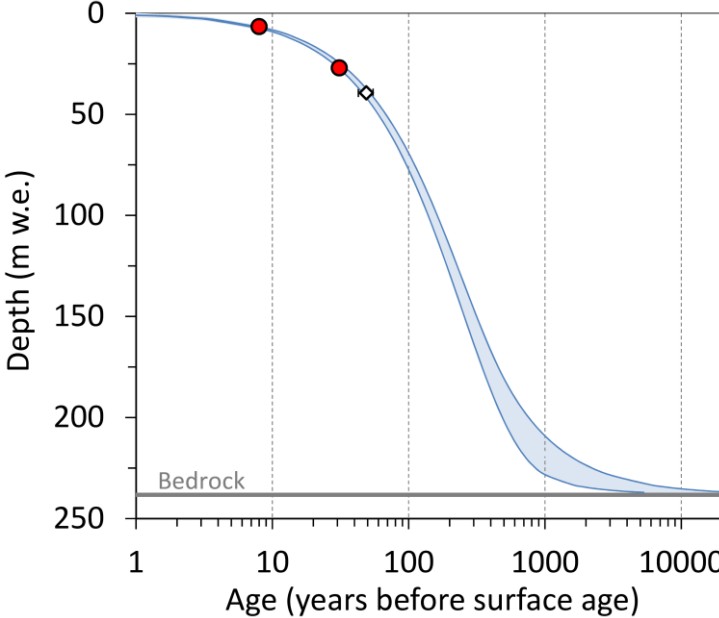

**Figure 6.** Model based estimate of the age-depth relationship down to bedrock for the ADA16 drill site. Red dots

show the 1986 and 1963 [137]Cs horizons used to fit the model. The estimated age for the bottom of the 46 m long ADA16 ice core is shown in addition (open diamond, not used for model tuning). The shaded area indicates the range of estimates as confined by the upper and lower uncertainty bounds (thin blue lines).

**5      Conclusions**

Thanks to a combination of methods we succeeded in building a reliable timescale for the 46 m deep Adamello ice core ADA16, using ALC based on pollen and rBC, as well as the radioactive decay of $^{210}$Pb and time markers identified by maxima in $^{137}$Cs. According to the chronology we propose, the ADA16 ice core covers a period of about 50 years from ~1944 to 1993 AD. For the period 1963-86 an average annual net accumulation rate of $0.9 \pm 0.03$ m w.e. a$^{-1}$ could be determined. On the other hand, no accumulation was preserved for about the last 20 years

at the Adamello 2016 drilling site on Pian di Neve (Italian Alps), as indicated by all approaches used to estimate the age of the surface, yielding an age older than the drilling date of 2016. This result is consistent with previous studies on mass balance in the region and suggests that the coring site is currently located below the ELA. The lack of retained accumulation across the former accumulation zone indicates a glacier that cannot survive (Pelto, 2010): the Adamello glacier is clearly at high risk under present-day conditions. Our results undoubtedly highlight

the fact that recent warming does not only compromise signal preservation, but can also lead to severe loss of glacier surface mass, even at high altitude. The removal of annual accumulation layers prevents the use of the drilling date as an anchor point for annual layer counting. Here, we demonstrated that establishing a chronology is nevertheless possible when using tracers in particulate form (pollen and spores, BC) or attached to particles ($^{210}$Pb), as they show to be least affected by melting. Finally, our results are encouraging for optimism that at least

some climatic and environmental signal could still be preserved, particularly in the deeper, potentially colder layers of the Adamello glacier. Those, according to an estimate based on an ice-flow modelling approach, potentially reach an age of several thousand years. In these terms, our study becomes relevant on a global scale, as it opens new perspective on studying temperate glaciers and their potential as environmental and climatic paleoarchives.

**6      Acknowledgements**

This work is a contribution to the project CALICE- Calibrating biodiversity in glacier ice, a multidisciplinary program between the University of Innsbruck, the Free University of Bozen – Bolzano and the Fondazione Edmund Mach in San Michele, funded by the EVTZ/Austrian Science Fund (IPN 57-B22). This is the CALICE-project publication no. 2. We would like to thank all member of the CALICE scientific consortium, especially those who helped during the coring activities and the processing of the ice core. We are grateful to the ENEA

drilling team and the alpine guide Nicola Viotti (Guide Alpine Valsusa) for their excellent work during the coring campaign. Drilling has been possible thanks to a specific grant (POLLice) to FEM (Fondazione Edmund Mach) from the Autonomous Province of Trento (PAT) and logistic support (helicopter flights) provided by Dr. Ernesto Sanutuliana. Eurocold Lab activities were partially funded by the Italian Regional Affair Ministry. We would like to thank also the MUSE-Museum of Science of Trento for its support, in particular Christian Casarotto and Elena

Bertoni. Finally, a special thanks goes to Marco Filipazzi and Giovanni Baccolo for their precious assistance in processing the ice core samples at the Eurocold Facility, to Silvia Köchli from PSI for $^{210}$Pb sample processing as well as to Dimitri Osmont, Anja Eichler, Sabina Brütsch and Susanne Haselbeck from PSI for rBC analysis.

**7      Authors contribution**

Daniela Festi performed pollen analyses on the ice and provided the pollen ALC together with Klaus Oeggl. Theo Jenk and Margit Schwikowski were responsible for BC and Pb-210 analyses and building the relative age-depth model. Valter Maggi was leader of the drilling campaign, coordinated ice core processing and cutting, and density measurements. All authors contributed actively writing the manuscript.

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
