# Peer review of "Significant mass loss in the accumulation area of the Adamello glacier indicated by the chronology of a 46 m ice core"

_The Cryosphere, 2020_

## Referee Comment (RC1) · Mauri Pelto (Referee) · 26 Dec 2020

Festi et al (2020) provide a detailed review of the dating and accumulation record revealed from an ice core in the former accumulation zone of the Adamello Glacier. The change from a net accumulation rate of ∼0.9 ma-1 to no preservation of accumulation is as important as the dating of the core. More attention needs to be given to other dated temperate glacier cores, in particular in the Alps. There are regional mass balance records that extend over at least part of the ice core period and the period when no accumulation has been preserved that can highlight the pattern identified here. Further records from this same glacier, also referred to as Mandrone Glacier, which are

more recent should be noted (Ranzi et al. 2010; Grossi et al. 2012) (1995-2009). The greater context will strengthen the findings of this paper.

14: Reword: "Dating glaciers is an arduous yet essential task in ice core studies, which becomes more challenging when the glacier is experiencing mass loss in the accumulation zone as result of climate warming leading to an older ice surface of an unknown age."

22: You have a short abstract and could add what is equally important to the ability to date this core, something like "The change in mass balance at the coring site, in the former accumulation zone, but which no longer retains accumulation, is in the range of $\sim$1 ma-1".

32: ".. even in what had formerly been the accumulation zone."

36: "... making annual layer counting impossible when the seasonality in the signal is lost"

37: Reference for the percolation issue for annual signal retention would be good.

39: Reword, "To date relatively few ice cores from temperate high elevation glaciers have been successfully be dated (von Gunten et al., 1982; Kang et al., 2015; Pavlova et al., 2015; Kaspari et al., 2020; Gäggeler et al., 2020)." This avoids having to be accurate in citing every dated ice core from an alpine glacier. Other examples I have had a chance to review from temperate glacier settings in North America alone include Naftz et al. (1996), Neff et al. (2017) and Yalcin et al. (2006). In the Alps you should refer to specific locations where this has been accomplished in addition to Silvretta Glacier. Should mention the Colle Gnifetti core from Monte Rosa (Schwikowski et al. 1999), and Col du Dom on Mont Blanc (De Angelis and Gaudichet, 1991).

47: State elevation for comparison to Ortles Glacier.

67: Reword "Adamello Glacier is located at a relative low elevation of 2500-3400 m a.s.l. (Figure 1) and currently affected by considerable mass loss (Maragno et al.,

2009) with recent negative mass balance observed even in the accumulation zone above ? m." This likely indicates the glacier does not have a persistent accumulation zone. What has been the ELA in recent years?

85: Figure 1 is not satisfactory. Figure 1 the left panel for Adamello Glacier is not sufficiently clear to be useful. The field area maps need to include elevation contours, longitude-latitude and scale, since these can easily be found in GLIMS or Grossi et al. (2012).

155: What is the timing of the potential multi-year pollen signal and does that coincide with years of high snowlines when snowcover was lost at glaciers with mass balance records? Review Huss et al (2015) and Carturan et al (2013), the latter in Table 3 also lists annual ELA.

188: Any insight on why the usual decrease in activity with depth was not observed?

200: Can you quantify very close agreement?

205: The year 1998 also marks the beginning of a periods of substantially more negative mass balance in the region Huss et al. (2015) and Carturan et al. (2013). Relate to mass balance observations on Adamello (Mandrone) Glacier for part of the period where a record is not retained Ranzi et al. (2010) and Grossi et al. (2012).

222: It is worth quantifying the size of the pollen grains to the ice crystals. Does the lack of pollen migration suggest the pollen is incorporated in ice crystals, or that meltwater percolation rates are too low to mobilize? You may not have insight on this, but if you do it will be interesting.

229: Reword, because it more accurate to say no accumulation has been retained. "Based on the good agreement and our confidence in the dating we can conclude that for at least two decades no net accumulation has been preserved at the drill site."

239: The annual accumulation that had existed 1963-1986 indicates that mass balance in this area of the accumulation since 1998 when accumulation is not preserved has

declined by more than 1 m on average. This is as important a finding as the dating and should be emphasized more.

255: Explain why this model is a good choice and how it has worked in a similar environment. How does this compare to methods used at Colle Gniffeti by Lüthi and Funk (2000).

274: "..indicating no accumulation preserved during the last 20 years." The lack of retained accumulation across an accumulation zone also indicates a glacier that cannot survive (Pelto, 2010).

References

Carturan, L., Baroni, C., Becker, M., Bellin, A., Cainelli, O., Carton, A., Casarotto, C., Dalla Fontana, G., Godio, A., Martinelli, T., Salvatore, M. C., and Seppi, R.: Decay of a long-term monitored glacier: Careser Glacier (Ortles-Cevedale, European Alps), The Cryosphere, 7, 1819–1838, https://doi.org/10.5194/tc-7-1819-2013, 2013.

De Angehs, M. and Gaudichet, A.: Saharan dust deposition over Mont Blanc (French Alps) during the last 30 years. Tellus, 43B(1), 61–75, 1991.

Grossi, G.; Caronna, P.; Ranzi, R.: Hydrologic vulnerability to climate change of the Mandrone glacier (Adamello-Presanella group, Italian Alps). Adv. Water Resour. 2013, 55, 190–203, 2010.

Huss, M., Dhulst, L., and Bauder, A.: New long-term mass-balance series for the Swiss Alps. Journal of Glaciology, 61(227), 551-562. doi:10.3189/2015JoG15J015, 2015.

Lüthi, M.,and Funk, M.: Dating ice cores from a high Alpine glacier with a flow model for cold firn. Annals of Glaciology, 31, 69-79. doi:10.3189/172756400781820381, 2000.

Naftz, D. L., Klusman, R. W., Michel, R.L., Schuster, P.F., Reddy, M.M., Taylor, H.E., Yanosky, T.W. and McConnaughey, E.A.: Little Ice Age evidence from a south‐central North American ice core, U.S.A. Arctic Alpine Res., 28, 35–41, 1996.

Neff, P., Steig, E., Clark, D., McConnell, J., Pettit, E., and Menounos, B.: Ice-core net snow accumulation and seasonal snow chemistry at a temperate-glacier site: Mount Waddington, southwest British Columbia, Canada. Journal of Glaciology, 58(212), 1165-1175. doi:10.3189/2012JoG12J078, 2012.

Pelto, M. S.: Forecasting temperate alpine glacier survival from accumulation zone observations, The Cryosphere, 4, 67–75, https://doi.org/10.5194/tc-4-67-2010, 2010.

Ranzi, R., Grossi, G., Gitti, A. and Taschner, S.: Energy and mass balance of the Mandrone glacier (Adamello, Central Alps). Geogr Fis Din Quat 2010;33:45–60, 2010.

Schwikowski, M., Brutsch, S., Gaggeler, H. and Schotterer, U.. 1999. A high-resolution air chemistry record from an Alpine ice core: Fiescherhorn glacier, Swiss Alps. J. Geophys. Res., 104(D11), 13,709–13,719, 1999.

Yalcin, K., Wake, C., Kreutz, K. and Whitlow, S.: A 1000-yr record of forest fire activity from Eclipse Icefield, Yukon, Canada. Holocene, 16, 200–209, 2006.
* * *

---

## Referee Comment (RC2) · Roberta Pini (Referee) · 9 Mar 2021

The paper by Festi et al. assesses the chronology if the ADA16 ice core drilled at 3100 m asl at Pian di Neve (Adamello Glacier). The chronological approach is based on the comparison of three independent dating methods and their lines of evidence, namely peaks in biological proxies concentration (palynomorphs and refractory BC), 137Cs and 210Pb geochronometry. Methods and results are correctly presented. Here below I list some points that need to be considered by the authors for an improvement of the manuscript (text + figures).

10. Dipartimento di Scienze dell'Ambiente e della Terra, Università Milano Bicocca 99.

how many of the 536 samples taken for palynology were actually analyzed? Looking at Fig. 2, it seems that they are way less than 536. 147-151: the information represented in the PCA plot seem to be important for the interpretation of the pollen signal stored in the ADA 16 ice core. Please add the PCA plot in the main text. 152: can you determine the time length of the multiple year signal condensed at 2.1 and 12.2 m w.e. equivalent? can pollen concentration help with this issue? 217: "The dating of the three independent dating methods ...". Please rephrase. 221: is it just pollen or pollen+spores? if so, use the term palynomorphs 295: Filipazzi instead of Filippazzi

Fig. 1: please add lat-long grids to the insets showing images of glaciers and surrounding mountains and some geographic names to help the readers in localizing the site

---

## Referee Comment (RC3) · Pascal Bohleber (Referee) · 16 Mar 2021

General comments

Festi et al. present chronological information for a 46 m temperate ice core drilled at Pian di Neve, Adamello glacier. The ice core was dated through a novel combination of pollen and refractory black carbon analyses alongside with radiometric dating by 210Pb and already existing 137Cs horizons. By this means, the authors are able to constrain the age of the surface at the time of drilling, which remained unknown due to existing evidence of prolonged negative mass balance at the site. This is addressing an
issue of broad relevance to ongoing and future drilling efforts aiming to recover valuable environmental and climatic records at sites that already undergo ice loss at the surface due to persisting warming conditions. I find the manuscript interesting, well-written and suitable for The Cryosphere. I also have a few comments and suggestions on how to improve the manuscript.

I find the new approach to constrain the surface age and to derive an average value for the former net snow accumulation to be the key deliverable of the manuscript. This is of interest not only for the dating of ice cores but provides also important overlap with glaciological investigations at the site, in particular regarding the mass balance reconstruction. This latter point certainly provides additional value to the manuscript and should deserve some more emphasis. Possible additions could be made to the discussion part and in the abstract. For instance, the new evidence for a surface dating to 1995 presented here appears to be nicely consistent with the mass balance investigation by Ranzi et al. (2010), which shows a persistent negative net mass balance since 1995 (one exception 2001). In their Table 1, Ranzi et al. (2010) also provide seasonal information on mass balance that may be interesting to take into account with regards to the pollen seasonal signal. It may also be worth pointing out that such point mass balance reconstructions have particular value as they have been shown to reflect changes in climate better than total mass balance or terminus fluctuation (Vincent et al., 2017). Relatedly, it has also been shown that point mass balance changes can reveal clear regional consistencies, which is interesting to note in the framework of the comparison with Ortles and Silvretta (lines 234).

Specific comments

To aid a better comparison with the existing glaciological datasets, Figure 1 should contain a better map of the drilling area, including at least some topographical detail and preferably contour lines. At present, very little can be learned about the position of the drilling site. For instance, it seems like several catchment areas may exist for the deeper ice core sections.

The glaciological setting also concerns another important aspect: It is stated that the core was drilled at the location of greatest ice thickness (line 73). It was not possible for me to verify this statement, however. The seismic campaign of Picotti et al. (2017) focused on one profile. The ground-penetrating radar survey seems to originate in Frassoni et al. (2001), but in spite of making a serious effort, I was unable to retrieve this paper. Ideally an ice thickness map could be added to Figure 1.

The ice thickness information could also aid in section 4.1 concerned with an age extrapolation to bedrock. The results are interpreted here basically as reconnaissance for a potential new drilling effort targeting to reach bedrock. I appreciate that the authors openly state that the use of the Dansgaard-Johnsen (D-J) model serves to make merely a crude estimate (line 259). This is not just due to the constraints located only in the upper third depth range, however. Here the clarification of additional points helps to put the inferred maximum age range into context: First, regarding the assumption that the ice is frozen to bedrock – how likely is this given the present evidence? Second, it is reported that the ice thickness value was determined by ground-penetrating radar, but this could have been via seismics instead? (line 251, citing Picotti et al., 2017). Regardless, the ice thickness value will have considerable uncertainty and the calculated dating function is typically sensitive to this. Therefore, a simple sensitivity study using the maximum vs minimum in ice thickness range would provide a more realistic insight regarding the age range expected from this estimation. This could be added as an illustration to Figure 5, which shows a 95% confidence interval but lacks detail on how this was derived.

Technical comments

Line 21: "... mass loss affecting this glacier even in the accumulation zone". Since mass loss is persistent today, it might be better to say "former accumulation zone", including at other instances in the text.

Line 21: "we show that it is possible to obtain a reliable timescale for such a temperate

glacier". This has been shown before. I would suggest to emphasize more the novelty of this work in the abstract, specifically regarding the combination of pollen and rBC and the resulting constraints for the surface age.

Line 27: Maybe say "regional scale"?

Line 33: See comment on accumulation zone above.

Line 37: This is of course very important. Maybe use one of the following citations here to back up this statement?

Line 81: "wet conditions" – what do you mean? What kind of problem stopped the drilling?

Line 142: Could the striking synchronicity between pollen maxima and rBC be quantified somehow, e.g. through a correlation measure? Out of curiosity, can you make out different regimes if the two datasets are used in a scatterplot? This could help to detect, for instance, anomalously high pollen or rBC values.

Figure 2: Personally I would find a zoom-in into a smaller depth interval of added value here.

Line 230: Delete "over the past years"

Line 279: I suggest to rephrase this statement considering that the results comprise pollen and rBC and the upper 46 m. It remains to be shown if a climatic and environmental signal, e.g. in the chemical impurities and stable water isotopes is preserved at the site, including the deep ice layers.

References

Frassoni, A., Rossi, G. C., & Tamburini, A. (2001). Studio del ghiacciaio dell'Adamello mediante indagini georadar. Suppl Geogr Fis Din Quat, 5, 77-84.

Picotti, S., Francese, R., Giorgi, M., Pettenati, F., & Carcione, J. M. (2017). Estimation

of glacier thicknesses and basal properties using the horizontal-to-vertical component spectral ratio (HVSR) technique from passive seismic data. Journal of Glaciology, 63(238), 229-248.

Ranzi, R., Grossi, G., Gitti, A., & Taschner, S. (2010). Energy and mass balance of the mandrone glacier (Adamello, Central Alps). Geografia Fisica e Dinamica Quaternaria, 33(1), 45-60.

Vincent, C., Fischer, A., Mayer, C., Bauder, A., Galos, S. P., Funk, M., ... & Huss, M. (2017). Common climatic signal from glaciers in the European Alps over the last 50 years. Geophysical Research Letters, 44(3), 1376-1383.

---

## Referee Comment (RC4) · Anonymous Referee #4 · 10 Apr 2021

The paper presents new data about the accumulation/ablation rate of the Adamello Glacier, the largest in Italy, estimated from a new ice core. Results are interesting, however some weaknesses need some improvements.

Abstract

It seems there is a contraddiction when stating that the surface is clearly old and that the drilling is in the accumulation zone. Scientific literature about the Adamello glacier mass balance indicates that the area is not in the accumulation zone.

Line 45. I am quite surprised by this conclusion. Being the altitude at Pian di Neve 759

m below the Alto Ortles Glacier we expect a 4°C-5°C mean temperature below and so definitely stronger temperate glacier conditions than Ortles.

Maragno et al. 2009 indicated an area loss of 19% and not a mass loss in the period 1983-2003. A more precise description of the meteorological and mass balance context is recommended also based on a more complete literature review of mass balance in the region.

Line 126 Because of the melting conditions at the surface I ask to comment how the exact timing of the radionuclides can be ensured. I have doubts about the correspondence between ice core depth and age.

Figure 2. I do not see a clear correspondence between Pollen&Spores and rBC in Figure 2A and 2B if any was expected. The timing seems to be fairly kept but the correlation seems to be very weak. Can the authors plot a scatter plot with the two variables.

Figure 3 shows a fair correspondence. Can the Authors plot a moving average line to better identify the peaks in 210Pb at Silvretta and Adamello?

Figure 5 is quite problematic. With just three points in the 1-40 years range it seems difficult to fit the Dansgaad Johnsen flow model up to 10000 years also considering the morphology of the bedrock underneath Pian di Neve. So I agree with the Author's comment at line 260-261. I would ad 'very crude'.

Conclusions. In the conclusions I would better stress the estimated accumulation rate of 0.8-0.9 m w.e. yr-1 which is quite convincing than the Dansgaard-Johnsen model age estimate which is very uncertain.

---

## Author Comment (AC1) · 10 Jun 2021

**Reply to Referee #1 Mauri Pelto**

**Referee #1** Festi et al (2020) provide a detailed review of the dating and accumulation record revealed from an ice core in the former accumulation zone of the Adamello Glacier. The change from a net accumulation rate of ~0.9 ma-1 to no preservation of accumulation is as important as the dating of the core. More attention needs to be given to other dated temperate glacier cores, in particular in the Alps. There are regional mass balance records that extend over at least part of the ice core period and the period when no accumulation has been preserved that can highlight the pattern identified here. Further records from this same glacier, also referred to as Mandrone Glacier, which are more recent should be noted (Ranzi et al. 2010; Grossi et al. 2012) (1995-2009). The greater context will strengthen the findings of this paper.

**Authors:** We thank referee #1, Mauri Pelto, for the useful suggestions to improve our manuscript and we here address his recommendations.

**Referee #1** 14: Reword: "Dating glaciers is an arduous yet essential task in ice core studies, which becomes more challenging when the glacier is experiencing mass loss in the accumulation zone as result of climate warming leading to an older ice surface of an unknown age."

**Authors:** Done.

**Referee #1** 22: You have a short abstract and could add what is equally important to the ability to date this core, something like "The change in mass balance at the coring site, in the former accumulation zone, but which no longer retains accumulation, is in the range of ~1 ma-1".

**Authors:** Sentences now reads: "For the period of 1995-2016 the mass balance at the drilling site (former accumulation zone) decreased on average of about 1 m w.e. a-1 compared to the period 1963-1986."

**Referee #1** 32: ".. even in what had formerly been the accumulation zone."

**Authors:** Done.

**Referee #1** 36: "... making annual layer counting impossible when the seasonality in the signal is lost"

**Authors:** Done.

**Referee #1** 37: Reference for the percolation issue for annual signal retention would be good.

**Authors:** Reference has been added.

**Referee #1** 39: Reword, "To date relatively few ice cores from temperate high elevation glaciers have been successfully be dated (von Gunten et al., 1982; Kang et al., 2015; Pavlova et al., 2015; Kaspari et al., 2020; Gäggeler et al., 2020)." This avoids having to be accurate in citing every dated ice core from an alpine glacier.

**Authors:** Changed accordingly.

**Referee #1** Other examples I have had a chance to review from temperate glacier settings in North America alone include Naftz et al. (1996), Neff et al. (2017) and Yalcin et al. (2006).

**Authors:** We included Naftz et al. (1996), Neff et al. (2012, this is the correct date) but not Yalcin et al (2006) because it does not mention the fact that it's a temperate glacier.

**Referee #1** In the Alps you should refer to specific locations where this has been accomplished in addition to Silvretta Glacier. Should mention the Colle Gnifetti core from Monte Rosa (Schwikowski et al. 1999), and Col du Dom on Mont Blanc (De Angelis and Gaudichet, 1991).

**Authors:** These references refer to cold glaciers, not temperate, and have therefore not been included.

**Referee #1** 47: State elevation for comparison to Ortles Glacier.

**Authors:** Added.

**Referee #1** 67: Reword "Adamello Glacier is located at a relative low elevation of 2500-3400 m a.s.l. (Figure 1) and currently affected by considerable mass loss (Maragno et al., 2009) with recent negative mass balance observed even in the accumulation zone above?" This likely indicates the glacier does not have a persistent accumulation zone. What has been the ELA in recent years?

**Authors:** To avoid the repetition of "Adamello" at the beginning of two consecutive sentences we changes into" The Adamello is the largest glacier in Italy with an extension of 16,3 km$^2$ (Smiraglia and Diolaiuti, 2015) and being located at a relative low elevation of 2500-3400 m a.s.l. (Figure 1) it is currently affected by considerable area loss…". Yes, the glacier does not have a persistent accumulation zone. This consideration has now been added. Direct information about the ELA are lacking for the site but according to Žebre et al (2021) the Adamello coring site is likely located below the current ELA. We added this information in the site description, in the "Age of surface" and conclusion paragraph.

**Referee #1** 85: Figure 1 is not satisfactory. Figure 1 the left panel for Adamello Glacier is not sufficiently clear to be useful. The field area maps need to include elevation contours, longitude-latitude and scale, since these can easily be found in GLIMS or Grossi et al. (2012).

**Authors:** Figure 1 has been modified**:**

[Figure]

**Figure 1**. Map showing the locations of the Adamello (red diamond) and Silvretta (light-blue diamond) Glaciers) and respective zoom-in maps on ice core drilling sites: Adamello (red star); Silvretta (blue star) . All maps are north-up oriented.

**Referee #1** 155: What is the timing of the potential multi-year pollen signal and does that coincide with years of high snowlines when snowcover was lost at glaciers with mass balance records? Review Huss et al (2015) and Carturan et al (2013), the latter in Table 3 also lists annual ELA.

**Authors:** the timing is 1977-79 and 1989-91. There is no striking coincidence between these years and particularly negative mass balance years in the records suggested. This might be due to regional and local variability.

**Referee #1** 188: Any insight on why the usual decrease in activity with depth was not observed?

**Authors:** Possible reasons could be related to changes in the seasonal distribution of annual precipitation/deposition rates, or to changes in the main source origin of recorded air masses (lower [210]Pb activity concentration over the oceans compared to coastal or continental sites). For more details, see Gäggeler et al., 2020. Because we can currently only speculate, these potential explanations were not included in the manuscript. However, we now added further information explaining what requirements need to be fulfilled in order to observe the typical exponential decrease in 210Pb concentration activity with depth.

**Referee #1** 200: Can you quantify very close agreement?
**Authors:** Within a few years. As can be seen in Figure 5 the dating by the two approaches in the period 1963-86 is lying on top of each other. That this is particularly the case for the period 1963-86 and "within a few years" was now added to the text now also including a reference to Fig. 5.

**Referee #1** 205: The year 1998 also marks the beginning of a periods of substantially more negative mass balance in the region Huss et al. (2015) and Carturan et al. (2013). Relate to mass balance observations on Adamello (Mandrone) Glacier for part of the period where a record is not retained Ranzi et al. (2010) and Grossi et al. (2012).
**Authors:** We now added this considerations in the paragraph "Final Adamello chronology" relating to the age of surface.

**Referee #1** 222: It is worth quantifying the size of the pollen grains to the ice crystals. Does the lack of pollen migration suggest the pollen is incorporated in ice crystals, or that meltwater percolation rates are too low to mobilize? You may not have insight on this, but if you do it will be interesting.
**Authors:** Unfortunately, we don't have a clear insight on this topic and therefore every hypothesis would be speculative. Clearly, we agree that this is an issue worth investigating.

**Referee #1** 229: Reword, because it more accurate to say no accumulation has been retained. "Based on the good agreement and our confidence in the dating we can conclude that for at least two decades no net accumulation has been preserved at the drill site."
**Authors:** Changed accordingly, except we prefer to keep "20 years" instead of "two decades".

**Referee #1** 239: The annual accumulation that had existed 1963-1986 indicates that mass balance in this area of the accumulation since 1998 when accumulation is not preserved has declined by more than 1 m on average. This is as important a finding as the dating and should be emphasized more.
**Authors:** We agree and thank the reviewer for this input. We now stress this finding more and added/reworded the related section in the conclusion: "For the period 1963-86 an average annual net accumulation rate of $0.9 \pm 0.03$ m w.e. $a^{-1}$ could be determined. On the other hand, no accumulation was preserved for about the last 20 years at the Adamello 2016 drilling site on Pian di Neve (Italian Alps) as indicated by all approaches used to estimate the age of the surface, yielding an age older than the drilling date of 2016."

**Referee #1** 255: Explain why this model is a good choice and how it has worked in a similar environment. How does this compare to methods used at Colle Gnifetti by Lüthi and Funk (2000).
**Authors:** The Dansgaard-Johnson model is a standard 1D ice-flow model, in the past being widely applied both on polar ice sheets and alpine glaciers. It is a slightly more sophisticated version of the maybe more famous Nye model, considering a non-uniform vertical strain rate, i.e. a horizontal shear zone layer in the bottom part. Based on borehole deformation measurements performed on a variety of alpine glaciers, this certainly yields a more realistic numerical representation of reality. In case of basal sliding, the horizontal shear stress decreases with the sliding velocity. This can be taken into account in this model (our estimate is based on a glacier frozen to bedrock, i.e. no sliding assumed based on the findings by Picotti et al., 2017 of an absence of melt water at the base). While the Dansgaard-Johnson model used here is a 1D model, the modelling approach used/developed by Lüthi and Funk (2000) is based on much more complex 2D and 3D models (also including improvements in modelling the firn part which is irrelevant here because no firn layer exists and more), requiring extensive additional observational data (e.g. digital elevation maps of the surface and the bedrock, radio-echo soundings of the ice thickness, firn density and the englacial temperature fields either prescribed or calculated in coupled models, measured surface velocities, density profiles, the ages of chemically dated layers in ice cores and borehole closure measurements). Obviously, the approach by Luthi and Funk (or comparable 3D approaches) is much more costly in terms of time,

measurements and computation. If the purpose is solely about estimating the age of ice at a certain depth at an ice core drill site (typically in the accumulation zone, selected specifically in an area of least complicated ice flow), then for the rather small alpine glaciers with complex glacier/bedrock geometries, even the most complex models will yield high uncertainties in their age-depth estimates. Also, they may still not yield results in (close) agreement with observations, i.e. the age of absolutely dated horizons (e.g. Licciulli et al., 2020). This is true, unless they can be constrained by actual dated horizons available throughout the core. If not, any such model can only provide a best guess associated with rather large uncertainties particularly for the bottom few meters above bedrock (certainly true also for 2D and 3D in the absence of constraint from borehole measurements, such as temperature and deformation like in our case).

In the revised version we now better and more carefully explain the limitations of age modelling, remaining cautious in how to portray our interpretation of results and with a clear main message in which we have high confidence in its robustness even if considering all uncertainties. See also related comments and answers to the other referees. The section now reads:

"For an estimation of the potential age range accessible by the Adamello ice archive, the one-dimensional Dansgaard-Johnsen ice-flow model was applied (Dansgaard and Johnsen, 1969). For the resulting age-depth relationship estimate shown in Figure 6, model parameters were as follows. Based on the bedrock depth determined by Picotti et al. (2017) using seismic measurements, the value for glacier thickness at the drill site was $265 \pm 5$ m ($238 \pm 4.5$ m w.e.). The bottom shear zone thickness was assumed to be 15 % of the glacier thickness. This is slightly lower than the ~20 % typically observed for cold and polythermal high-elevation glaciers (e.g. Jenk et al., 2009; Uglietti et al., 2016; Gabrielli et al., 2016; Licciulli et al., 2020) but likely more reasonable for a temperate glacier (e.g. Kaspari et al., 2020). In any case, because constraining information from dated age horizons is lacking for the bottom part, a relatively large uncertainty of $\pm 10$ % was assigned. With these parameter settings, the value for the annual accumulation rate was found by tuning for a best model-fit to the dated 1986 and 1963 $^{137}$Cs horizons (least squares approach). The dating uncertainty and the uncertainties associated with the pre-set model parameters described above were employed to derive upper and lower bound estimates (to transfer the contribution form uncertainty in ice thickness to uncertainty in age, relative depths were used).

The model - nicely matching the determined bottom age for the ADA16 core and accounting for layer thinning (vertical strain) – provides us a best estimate of the mean annual accumulation rate at the ADA16 drill site for the period ~1946 to 1986 of $0.9 \pm 0.03$ m w.e. $a^{-1}$. However, the assumption of steady-state conditions and the complexity of bedrock geometry and glacial flow in the deepest part of high-alpine glaciers strongly limits a realistic modelling of strain rates (and thus age) for the deeper parts, even using the most complex glaciological 3D ice-flow models. In our case, the lack of data for additional constraint in the deeper/older part, the assumption of steady-state conditions in annual accumulation rates (equal to an average value for the entire period contained in the archive) which are further based on a relatively short time range covered by the 46 m core only, the derived model-based age-depth relationship can only yield a current best estimate. Anyhow, this is at least sufficient to reveal the potential of the site. The Adamello ice archive is very likely to cover the last 1000 years. Being contained in the major part of the total ice thickness (about the upper 240 m of ice; ~220 m w.e.), a millennial-long record should thus be accessible in high resolution. Also, there is reasonable likelihood for a few thousand more years contained in the remaining ~10 % of ice below."

[Figure]

**Figure 6.** Model based estimate of the age-depth relationship down to bedrock for the ADA16 drill site. Red dots show the 1986 and 1963 [137]Cs horizons used to fit the model. The estimated age for the bottom of the 46 m long ADA16 ice core is shown in addition (open diamond, not used for model tuning). The shaded area indicates the range of estimates as confined by the upper and lower uncertainty bounds (thin blue lines).

**Referee #1** 274: "..indicating no accumulation preserved during the last 20 years." The lack of retained accumulation across an accumulation zone also indicates a glacier that cannot survive (Pelto, 2010).
**Authors:** We rephrased and added the suggested consideration and citation.

**References not included in the manuscript**

Žebre, M., Colucci, R. R., Giorgi, F., Glasser, N. F., Racoviteanu, A. E., & Del Gobbo, C.: 200 years of equilibrium-line altitude variability across the European Alps (1901−2100). *Climate Dynamics*, *56*(3–4), 1183–1201. https://doi.org/10.1007/s00382-020-05525-7, 2021.

---

## Author Comment (AC2) · 10 Jun 2021

**Reply to referee #2 Roberta Pini**

**Referee #2** The paper by Festi et al. assesses the chronology if the ADA16 ice core drilled at 3100 m asl at Pian di Neve (Adamello Glacier). The chronological approach is based on the comparison of three independent dating methods and their lines of evidence, namely peaks in biological proxies concentration (palynomorphs and refractory BC), 137Cs and 210Pb geochronometry. Methods and results are correctly presented. Here below I list some points that need to be considered by the authors for an improvement of the manuscript (text + figures).

**Authors**: We thank Roberta Pini for her useful suggestions to improve our manuscript, and we address as follows her the points of discussions.

**Referee #2:** how many of the 536 samples taken for palynology were actually analyzed? Looking at Fig. 2, it seems that they are way less than 536.

**Authors:** All of them were analysed and their entire content was quantified and identified but the large majority of the grains concentrated in those high concentration layers that are therefore visible in Figure 2.

**Referee #2** 147-151: the information represented in the PCA plot seem to be important for the interpretation of the pollen signal stored in the ADA 16 ice core. Please add the PCA plot in the main text.

**Authors:** Given that PCA results showed that 96% of the variance is included in the first principle component, a plot is not very readable and also a table of the component scores adds little value to the results. We here report the Table showing that all taxa correlate with the first principle component. Since it does not add value to the interpretation (i.e. possibility of extraction of a sub-annual pollen signal) we will not add it the manuscript.

**Table**  Component loadings of the first three Principal Components (PC) based on pollen concentration data in the Adamello ADA16 core.

| Taxa | 1 | 2 | 3 | Taxa | 1 | 2 | 3 |
|---|---|---|---|---|---|---|---|
| | **Components** | | | | **Components** | | |
| Chenopodiaceae T. | 1.000 | -0.016 | 0.002 | *Artemisia* | 0.999 | -0.021 | 0.013 |
| Brassicaceae | 1.000 | -0.017 | -0.010 | *Ulmus* | 0.999 | -0.035 | 0.014 |
| Asteraceae | 1.000 | -0.019 | 0.007 | *Rumex acetosella* | 0.999 | -0.035 | 0.012 |
| *Juglans* | 1.000 | -0.018 | 0.012 | Caryophyllaceae | 0.999 | -0.036 | 0.012 |
| Apiaceae | 1.000 | -0.025 | 0.000 | *Carpinus betulus* | 0.999 | -0.033 | 0.020 |
| *Tilia* | 1.000 | -0.024 | 0.008 | *Thalictrum* | 0.999 | -0.037 | 0.011 |
| *Plantago alpina* T. | 1.000 | -0.018 | -0.012 | *Ambrosia* | 0.999 | -0.013 | 0.034 |
| *Plantago lanceolata* T. | 1.000 | -0.026 | 0.000 | Cyperaceae | 0.999 | -0.035 | 0.014 |
| *Urtica* | 1.000 | -0.013 | 0.004 | Trilete spores | 0.999 | -0.038 | 0.016 |
| Ranunculaceae | 1.000 | -0.027 | 0.002 | *Calluna vulgaris* | 0.999 | -0.038 | 0.017 |
| *Pinus cembra* | 1.000 | -0.020 | 0.011 | *Salix* | 0.999 | -0.038 | 0.016 |
| Rumex acetosa T. | 1.000 | -0.028 | 0.001 | *Juniperus* | 0.999 | -0.040 | 0.016 |
| *Abies* | 1.000 | -0.022 | 0.017 | *Betula* | 0.999 | -0.023 | -0.001 |
| *Ephedra fragilis* T. | 1.000 | -0.026 | -0.003 | *Corylus avellana* | 0.999 | 0.011 | 0.006 |
| Cerealia | 1.000 | -0.023 | 0.012 | Monolete spores | 0.999 | 0.002 | 0.030 |
| Cannabaceae | 1.000 | -0.029 | 0.005 | *Fraxinus excelsior* | 0.999 | -0.012 | 0.004 |
| *Fagus* | 0.999 | -0.028 | 0.013 | *Ostrya* T. | 0.998 | 0.006 | 0.020 |
| Scrophulariaceae | 0.999 | -0.031 | 0.004 | *Quercus robur* T. | 0.998 | 0.009 | -0.036 |
| Rosaceae | 0.999 | -0.030 | 0.006 | *Alnus* | 0.995 | 0.030 | -0.022 |
| Cichoriaceae | 0.999 | -0.031 | 0.011 | *Olea* | 0.994 | 0.042 | -0.069 |
| *Saxifraga granulata* T. | 0.999 | -0.032 | 0.007 | *Castanea sativa* | 0.988 | 0.059 | -0.112 |
| *Saxifraga stellaris* T. | 0.999 | -0.032 | 0.006 | Gramineae | 0.986 | 0.125 | -0.095 |
| *Larix* | 0.999 | -0.017 | -0.001 | *Quercus ilex* T. | 0.982 | 0.059 | -0.121 |
| *Fraxinus ornus* | 0.999 | -0.005 | -0.017 | *Pinus* | 0.957 | 0.261 | 0.070 |
| Ericaceae | 0.999 | -0.030 | 0.018 | *Alnus viridis* | 0.939 | 0.231 | -0.216 |
| *Zea mays* | 0.999 | -0.032 | 0.018 | *Picea* | 0.914 | 0.340 | 0.209 |

**Referee #2** 152: can you determine the time length of the multiple year signal condensed at 2.1 and 12.2 m w.e. equivalent? can pollen concentration help with this issue?

**Authors:** Partially. The higher concentration in pollen is a sign that these layers include multiple years, however giving the variability of the pollen concentration of the core it is not possible to calculate the exact number of years. For this reason, the number of years were assigned taking into account also the 1986 and 1963 time horizons.

**Referee #2** 217: "The dating of the three independent dating methods ...". Please rephrase.

**Authors:** Rephrased into "The dating obtained with the three independent methods (ALC, $^{210}$Pb, $^{137}$Cs) is in excellent agreement."

**Referee #2** 221: is it just pollen or pollen+spores? if so, use the term palynomorphs

**Authors:** It is pollen and spores indeed. Changed.

**Referee #2** 295: Filipazzi instead of Filippazzi

**Authors:** Corrected.

**Referee #2** Fig. 1: please add lat-long grids to the insets showing images of glaciers and surrounding mountains and some geographic names to help the readers in localizing the site.

**Authors:** The figure has been updated also accordingly to suggestions by other review and it now includes detailed maps of the glaciers and an overview map:

[Figure]

**Figure 1.** Map showing the locations of the Adamello (red diamond) and Silvretta (light-blue diamond) Glaciers) and respective zoom-in maps on ice core drilling sites: Adamello (red star); Silvretta (blue star). All maps are north-up oriented.

---

## Author Comment (AC3) · 10 Jun 2021

Dear Pascal Bohleber,

thank you for your review. We have responded to all of your comments in the attached document.

Please also note the supplement to this comment:
https://tc.copernicus.org/preprints/tc-2020-334/tc-2020-334-AC3-supplement.pdf
Suppl. Geogr. Fis. Dinam. Quat.
V (2001), 77-84, 4 figg

Alberto FRASSONI (*), GianCarlo ROSSI (**) & Andrea TAMBURINI (*)

**STUDIO DEL GHIACCIAIO DELL'ADAMELLO MEDIANTE INDAGINI GEORADAR**

ABSTRACT: Frassoni A., Rossi GC. & Tamburini A., *Ground Penetrating Radar investigations on the Adamello Glacier*. (IT ISSN 0391-9838, 2001).

The results of the investigations carried out in 1997-98 in the Adamello Glacier area are discussed in this paper. Aim of the study was mapping the glacier surface and bedrock in order to better understand the dynamics and calculate the volume of the glacier. Aerial photogrammetry and GPS-assisted ground penetrating radar surveys both from the surface and by helicopter were carried out. Data were processed in ARC/INFO environment. The overall volume of the glacier reaches the value of about 870 million cubic meters, with a maximum ice thickness higher than 240 m in the Pian di Neve area. A good fit with the results of the previous surveys, carried out in the upper part of the glacier from the beginning of the 60s, can be observed, confirming that the Adamello glacier can be considered the largest alpine glacier.

KEY WORDS: Adamello Glacier, GPR, GPS, DEM.

RIASSUNTO: Frassoni A., Rossi GC. & Tamburini A., *Studio del Ghiacciaio dell'Adamello mediante indagini georadar*. (IT ISSN 0391-9838, 2001).

Il presente lavoro illustra la metodologia ed i risultati delle indagini geofisiche condotte nell'estate 1997 e nei primi mesi del 1998, volte alla caratterizzazione geometrica e volumetrica del ghiacciaio dell'Adamello ed alla ricostruzione della morfologia del substrato roccioso. Le indagini sono state eseguite utilizzando un radar geofisico, georeferenziato mediante tecnica GPS, per un totale di circa 70 km di rilievi, parte a terra e parte trasportando la strumentazione in elicottero. I dati acquisiti, elaborati mediante procedure operanti in ambiente ARC/INFO, hanno reso possibile una valutazione del volume complessivo del ghiacciaio, unitamente alla ricostruzione della superficie del ghiacciaio e della loro distribuzione areale, che concordano sostanzialmente con i risultati delle indagini sismiche eseguite all'inizio degli anni '60. Il volume complessivo del ghiacciaio è risultato pari a circa 870 milioni di m³, con uno spessore massimo superiore a 240 metri nell'area del Pian di Neve. Le caratteristiche morfologiche della superficie del ghiacciaio e del sottostante substrato roccioso consentono di confermare le più recenti ipotesi classificative, che considerano il ghiacciaio dell'Adamello come un unico apparato glaciale.

TERMINI CHIAVE: Ghiacciaio dell'Adamello, Georadar, GPS, DEM.

(*) *ISMES S.p.A., via Pastrengo 9, 24068 Seriate (BG).*
(**) *via Montello 8, Noale (VE).*
*Si ringrazia il dott. L. Veronese, Direttore del Laboratorio Geotecnico del Servizio Geologico della Provincia Autonoma di Trento, per la disponibilità dimostrata e la collaborazione offerta durante l'esecuzione dei rilievi.*

**INTRODUZIONE**

Il Ghiacciaio dell'Adamello, attualmente considerato il più vasto apparato glaciale delle Alpi, costituisce la principale riserva idrica solida d'Italia. A lungo ritenuto suddiviso in più individui, è formato da un vasto altipiano, detto Pian di Neve, dal quale si dipartono numerose colate in direzione delle valli laterali. La più importante ed estesa è quella del Mandrone, che si sviluppa in direzione settentrionale occupando la parte sommitale della Val di Genova (Mercíai, 1921 e 1924; Catasto dei Ghiacciai Italiani, 1961). In particolare il Passo Adamè, in corrispondenza del quale veniva tradizionalmente posta una soglia rocciosa, era considerato un limite tra la porzione meridionale (Pian di Neve) e quella settentrionale (Ghiacciaio del Mandrone).

A partire dagli anni '60 l'area è stata oggetto di numerose campagne di indagine, eseguite con diverse tecniche geofisiche, aventi lo scopo di valutare gli spessori del ghiaccio e soprattutto di ricostruire la morfologia del substrato roccioso, per poter tracciare le principali direzioni di flusso del ghiaccio e quindi definire con maggior chiarezza i limiti tra i vari individui. Si ricordano in particolare il rilievo sismico a riflessione eseguito da Carabelli (1962) ed i successivi rilievi geoelettrici eseguiti nel 1991 dalla Società Geoplan per conto della Società Alpinisti Tridentini (Bonardi & *alii*, 1995). Entrambe le indagini citate sono state limitate allo studio dell'area del Pian di Neve e del Passo Adamè.

Il Ghiacciaio dell'Adamello è attualmente considerato un unico apparato, con una morfologia riconducibile a quella di un ghiacciaio di tipo scandinavo, caratterizzato da un vasto altipiano ghiacciato di notevole spessore, da cui si dipartono radialmente numerose lingue (AA.VV. - Ghiacciai in Lombardia, 1992; Bonardi & *alii*, 1995).

**SCHEMA DELLE INDAGINI**

Nell'Estate 1997 è stata avviata per conto di ENEL S.p.A. una nuova campagna di indagine, ultimata agli inizi dell'anno successivo. Lo studio, esteso a tutta l'area

**Fig. 1.**

**Supplement:**

**Reply to Referee #3 Pascal Bohleber**

**Referee #3** General comments

Festi et al. present chronological information for a 46 m temperate ice core drilled at Pian di Neve, Adamello glacier. The ice core was dated through a novel combination of pollen and refractory black carbon analyses alongside with radiometric dating by 210Pb and already existing 137Cs horizons. By this means, the authors are able to constrain the age of the surface at the time of drilling, which remained unknown due to existing evidence of prolonged negative mass balance at the site. This is addressing an issue of broad relevance to ongoing and future drilling efforts aiming to recover valuable environmental and climatic records at sites that already undergo ice loss at the surface due to persisting warming conditions. I find the manuscript interesting, well-written and suitable for The Cryosphere. I also have a few comments and suggestions on how to improve the manuscript. I find the new approach to constrain the surface age and to derive an average value for the former net snow accumulation to be the key deliverable of the manuscript. This is of interest not only for the dating of ice cores but provides also important overlap with glaciological investigations at the site, in particular regarding the mass balance reconstruction. This latter point certainly provides additional value to the manuscript and should deserve some more emphasis.

**Authors:** We thank referee #3 Pascal Bohleber for his useful comments and suggestions to improve our manuscript and we hereby address the points of discussion.

**Referee #3** Possible additions could be made to the discussion part and in the abstract. For instance, the new evidence for a surface dating to 1995 presented here appears to be nicely consistent with the mass balance investigation by Ranzi et al. (2010), which shows a persistent negative net mass balance since 1995 (one exception 2001).

**Authors:** Considerations have been added in the discussion (Age of surface) and in the conclusions also including other mass balance records for the region.

**Referee #3** In their Table 1, Ranzi et al. (2010) also provide seasonal information on mass balance that may be interesting to take into account with regards to the pollen seasonal signal. It may also be worth pointing out that such point mass balance reconstructions have particular value as they have been shown to reflect changes in climate better than total mass balance or terminus fluctuation (Vincent et al., 2017). Relatedly, it has also been shown that point mass balance changes can reveal clear regional consistencies, which is interesting to note in the framework of the comparison with Ortles and Silvretta (lines 234).

**Authors:** We thank the reviewer for this valuable input regarding the value of point mass balance data. We included this information in the according section on the "Annual net accumulation rate for the period 1963-1986". Also, we agree that for the interpretation of pollen record information about seasonal mass balance are useful, but unfortunately Ranzi and al (2010) present data from the period 1995-2006 for which we don't have the corresponding layers.

**Referee #3** To aid a better comparison with the existing glaciological datasets, Figure 1 should contain a better map of the drilling area, including at least some topographical detail and preferably contour lines. At present, very little can be learned about the position of the drilling site. For instance, it seems like several catchment areas may exist for the deeper ice core sections.

**Authors:** Figure 1 has been improved following the suggestions provided by all reviewers.

**Referee #3** The glaciological setting also concerns another important aspect: It is stated that the core was drilled at the location of greatest ice thickness (line 73). It was not possible for me to verify this statement, however. The seismic campaign of Picotti et al. (2017) focused on one profile. The ground-penetrating radar survey seems to originate in Frassoni et al. (2001), but in spite of making a serious effort, I was unable to retrieve this paper. Ideally an ice thickness map could be added to Figure 1.

**Authors:** The position of the hole was defined after the survey of Picotti et al., (2017), who selected the seismic line based on the profiles by Frassoni et al. (2001) (file attached). Frassoni et al (2001) define the maximum depth with > 200 m (see fig. 2 therein), with reconstruction of the bedrock contours having a resolution of 25 m. Until now, no final map of the bedrock exists (not yet published) and such a figure can therefore unfortunately not be

included here. For the 46 m of core discussed here, knowledge about the precise ice thickness and bedrock topography is not relevant. It certainly will be for future discussion of results from a potential deep ice core from Adamello.

**Referee #3** The ice thickness information could also aid in section 4.1 concerned with an age extrapolation to bedrock. The results are interpreted here basically as reconnaissance for a potential new drilling effort targeting to reach bedrock.

**Authors:** Knowledge about the total ice thickness is certainly important for the age modelling. At this point we can only rely on the available data (see comment above). However, ice thickness is not the main uncertainty and not crucial to assess the potential of the site. The according section has been reformulated to better portray the main message there and now also more clearly discusses the limitations and uncertainties of these age estimates.

**Referee #3** I appreciate that the authors openly state that the use of the Dansgaard-Johnsen (D-J) model serves to make merely a crude estimate (line 259). This is not just due to the constraints located only in the upper third depth range, however. Here the clarification of additional points helps to put the inferred maximum age range into context: First, regarding the assumption that the ice is frozen to bedrock – how likely is this given the present evidence? Second, it is reported that the ice thickness value was determined by ground-penetrating radar, but this could have been via seismics instead? (line 251, citing Picotti et al., 2017). Regardless, the ice thickness value will have considerable uncertainty and the calculated dating function is typically sensitive to this. Therefore, a simple sensitivity study using the maximum vs minimum in ice thickness range would provide a more realistic insight regarding the age range expected from this estimation. This could be added as an illustration to Figure 5, which shows a 95% confidence interval but lacks detail on how this was derived.

**Authors:** The assumption that the ice is frozen to bedrock is the best we can do based on the present evidence (see Introduction). However, we cannot exclude the possibility for basal sliding. We use the age modelling only to assess the potential of the site. We reformulated the according section, being now more careful about describing the limitations of the modelling and degree of confidence one should assign to these modelled age estimates. The reviewer is correct, the thickness from Picotti et al., 2017 was derived by seismic data. Thank you for making us aware of this error, we corrected accordingly. We included the uncertainty in ice thickness to derive a more accurate estimate of the numerical model uncertainty (to convert the uncertainty contribution from the uncertainty in ice thickness, relative depths were used). The modelled age estimate is now presented as a band only, confined by the upper and lower estimate bounds (instead of the mean and a 95% confidence band). We now describe in more detail how uncertainties were derived. The section with Figure shown below now reads:

"For an estimation of the potential age range accessible by the Adamello ice archive, the one-dimensional Dansgaard-Johnsen ice-flow model was applied (Dansgaard and Johnsen, 1969). For the resulting age-depth relationship estimate shown in Figure 6, model parameters were as follows. Based on the bedrock depth determined by Picotti et al. (2017) using seismic measurements, the value for glacier thickness at the drill site was $265 \pm 5$ m ($238 \pm 4.5$ m w.e.). The bottom shear zone thickness was assumed to be 15 % of the glacier thickness. This is slightly lower than the ~20 % typically observed for cold and polythermal high-elevation glaciers (e.g. Jenk et al., 2009; Uglietti et al., 2016; Gabrielli et al., 2016; Licciulli et al., 2020) but likely more reasonable for a temperate glacier (e.g. Kaspari et al., 2020). In any case, because constraining information from dated age horizons is lacking for the bottom part, a relatively large uncertainty of $\pm10$ % was assigned. With these parameter settings, the value for the annual accumulation rate was found by tuning for a best model-fit to the dated 1986 and 1963 $^{137}$Cs horizons (least squares approach). The dating uncertainty and the uncertainties associated with the pre-set model parameters described above were employed to derive upper and lower bound estimates (to transfer the contribution form uncertainty in ice thickness to uncertainty in age, relative depths were used).

The model - nicely matching the determined bottom age for the ADA16 core and accounting for layer thinning (vertical strain) – provides us a best estimate of the mean annual accumulation rate at the ADA16 drill site for the period ~1946 to 1986 of $0.9 \pm 0.03$ m w.e. a$^{-1}$. However, the assumption of steady-state conditions and the complexity of bedrock geometry and glacial flow in the deepest part of high-alpine glaciers strongly limits a realistic modelling of strain rates (and thus age) for the deeper parts, even using the most complex glaciological 3D ice-flow models. In our case, the lack of data for additional constraint in the deeper/older part, the assumption of steady-state conditions in annual accumulation rates (equal to an average value for the entire period contained in the archive) which are further based on a relatively short time range covered by the 46 m core only, the derived model-based age-depth relationship can only yield a current best estimate. Anyhow, this is at least sufficient to

reveal the potential of the site. The Adamello ice archive is very likely to cover the last 1000 years. Being contained in the major part of the total ice thickness (about the upper 240 m of ice; ~220 m w.e.), a millennial-long record should thus be accessible in high resolution. Also, there is reasonable likelihood for a few thousand more years contained in the remaining ~10 % of ice below."

[Figure]

**Figure 6.** Model based estimate of the age-depth relationship down to bedrock for the ADA16 drill site. Red dots show the 1986 and 1963 [137]Cs horizons used to fit the model. The estimated age for the bottom of the 46 m long ADA16 ice core is shown in addition (open diamond, not used for model tuning). The shaded area indicates the range of estimates as confined by the upper and lower uncertainty bounds (thin blue lines).

*Technical comments*

**Referee #3** Line 21: ": : : mass loss affecting this glacier even in the accumulation zone". Since
mass loss is persistent today, it might be better to say "former accumulation zone",
including at other instances in the text.
**Authors:** Done

**Referee #3** Line 21: "we show that it is possible to obtain a reliable timescale for such a temperate glacier". This has been shown before. I would suggest to emphasize more the novelty
of this work in the abstract, specifically regarding the combination of pollen and rBC
and the resulting constraints for the surface age.
**Authors:** Done

**Referee #3** Line 27: Maybe say "regional scale"?
**Authors:** Done.

**Referee #3**
Line 37: This is of course very important. Maybe use one of the following citations here
to back up this statement?
**Authors:** Done. Reference has been added.

**Referee #3** Line 81: "wet conditions" – what do you mean? What kind of problem stopped the drilling?
**Authors:** Mechanical drilling under wet conditions (percolating surface melt water) causes technical problems, hindering the transport of drilling chips to the "chips barrel" because when becoming wet, they can clog the

transport spiral. We included this information now: "Drilling operations stopped at 46 m of depth due to wet conditions from percolation/inflow of surface melt water causing technical problems for mechanical drilling."

**Referee #3** Line 142: Could the striking synchronicity between pollen maxima and rBC be quantified somehow, e.g. through a correlation measure? Out of curiosity, can you make out different regimes if the two datasets are used in a scatterplot? This could help to detect, for instance, anomalously high pollen or rBC values.
**Authors:** The striking synchronicity in peak maxima is visible in Figure 2 and now very clearly in the newly added Figure 3 (referenced in the manuscript now). Because of different sampling resolution it is not possible to precisely quantify to what degree in terms of timing synchronicity exists, i.e. more precise than annual. In any case, a synchronicity in peak maxima between two parameters does not per se imply that there exists correlation between the two, which is also clearly not something we claim anywhere in the manuscript. Actually, we would not expect (high) correlation because of the different emission sources and processes for Pollen and BC (see comment to reviewer 4). Accordingly, a definition of what should be considered as "anomalously high" is not clear.

**Referee #3** Figure 2: Personally I would find a zoom-in into a smaller depth interval of added value here.
**Authors:** A figure (now figure 3) has been added showing a 5 m zoom into the record.

**Referee #3** Line 230: Delete "over the past years"
**Authors:** Done.

**Referee #3** Line 279: I suggest to rephrase this statement considering that the results comprise pollen and rBC and the upper 46 m. It remains to be shown if a climatic and environmental signal, e.g. in the chemical impurities and stable water isotopes is preserved at the site, including the deep ice layers.
**Authors:** We rephrased.

**References**
Frassoni A., Rossi G.C., Tamburini A.: Studio del Ghiacciaio dell'Adamello mediante georadar. Suppl. Geogr. Fis. Dinam. Quat. 4: 77-84, 2001.

---

## Author Comment (AC4) · 10 Jun 2021

**Reply to Anonymous Referee #4**

**Referee #4** The paper presents new data about the accumulation/ablation rate of the Adamello Glacier, the largest in Italy, estimated from a new ice core. Results are interesting, however some weaknesses need some improvements.

**Authors:** We thank anonymous referee #4 for the useful suggestions to improve our manuscript and as follows we address the recommendations.

**Referee #4** Abstract. It seems there is a contradiction when stating that the surface is clearly old and that the drilling is in the accumulation zone. Scientific literature about the Adamello glacier mass balance indicates that the area is not in the accumulation zone.

**Authors:** We rephrased into "former accumulation zone" to be more consistent.

**Referee #4** Line 45. I am quite surprised by this conclusion. Being the altitude at Pian di Neve 759 m below the Alto Ortles Glacier we expect a 4∘C-5∘C mean temperature below and so definitely stronger temperate glacier conditions than Ortles.

**Authors:** We agree that at Pian di Neve, compared to Ortles, stronger temperate glacier conditions are likely for the reasons pointed out by the reviewer. What we concluded about is the expected similarity in the trend between the two sites, not about them having the same temperature or even equal temperatures at the same depth. The trend we refer to is "temperatures around 0°C in the top part" (cannot be higher at Adamello because ice does not exists above 0°C) with temperatures of the ice being (at least) slightly lower below, thus a "cold deeper part". Of course, because of pressure due to the (large) ice thickness and potential geothermal heat, temperatures being higher again in the very bottom part cannot be excluded and will only be known once borehole temperatures in a deep core borehole will be available. Anyway, temperatures around 0°C in this part though would imply the presence of water there. This was however not confirmed by the analysis of Picotti et al. (2017).

To clarify, this section was changed. Thereby the reasoning pointed out by the reviewer was included: "With Pian di Neve (3100 m a.s.l.) being located in the same region, affected by similar climatic conditions, but with a far larger ice thickness, a similar trend in ice temperatures - the presence of temperate ice in the upper part and colder ice temperatures below - is not unlikely. While seismic analyses, do confirm the absence of melt water at the base of the glacier (Picotti et al., 2017), temperate ice conditions are however likely to exist to greater depth compared to the Alto dell'Ortles Glacier considering their difference in altitude."

**Referee #4** Line 45 Maragno et al. 2009 indicated an area loss of 19% and not a mass loss in the period 1983-2003. A more precise description of the meteorological and mass balance context is recommended also based on a more complete literature review of mass balance in the region.

**Authors:** "mass loss" has been corrected to "area loss". Additional information from and references about regional mass balance studies was included in the revised manuscript (see related comments/answers by/to the other referees).

**Referee #4** Line 126 Because of the melting conditions at the surface I ask to comment how the exact timing of the radionuclides can be ensured. I have doubts about the correspondence between ice core depth and age.

**Authors:** We are not entirely sure if we correctly understand the referees concerns here. The relation between the activity of radionuclides and time/age is given be the law of radioactive decay. We thus think the referee rather refers to the possibility of relocation of particles in the ice to greater depth by percolating melt water. However, if this is indeed the issue, we are a bit puzzled about this comment. Using different dating approaches to overcome the challenges imposed by post-depositional bias, as described and discussed in the manuscript to be undeniably present to some extent for each of the parameters used, is the strength and main message of our study. So to answer in short, in our study, the agreement between the independent dating using Pollen and rBC and the dating based on the radionuclides does argue against significant relocation of the radionuclides. That rBC, pollen and the radionuclides used are reasonably well preserved, i.e. not easily relocated or strongly affected in the presence of percolating meltwater is in agreement with findings of previous other studies already cited in the manuscript. Specifically for the radionuclides, we here would like to refer again to Gäggeler et al. 2020 ($^{210}$Pb) and for Pb and Cs to Avak et al., 2018 and Avak et al., 2019 who showed that these trace elements are reasonably well preserved in the ice in case of melt water percolation. The references to the studies by Avak et al. were now also added to the manuscript (in the Introduction).

**Referee #4** Figure 2. I do not see a clear correspondence between Pollen&Spores and rBC in Figure 2A and 2B if any was expected. The timing seems to be fairly kept but the correlation seems to be very weak. Can the authors plot a scatter plot with the two variables.

**Authors:** No correlation between Pollen&Spores and rBC is expected, at least not for the industrial period. While pollen and spores are of biogenic origin, BC (soot) is then to a large part of anthropogenic origin (see e.g. Sigl et al, 2018). Nothing about correlation, neither strong nor weak, is claimed in the manuscript (we even never used the word "correlation" or "correspondence"). Of relevance in the context of this study is only that for maxima and minima "The timing seems to be fairly kept" as the referee agrees on. In other words, what is important and the only point we make is the observation of synchronicity in pollen and spores and rBC peak maxima and minima (in the revised manuscript even better visible with the new Figure 3 added). This synchronism is mainly caused by vertical transport (stronger in spring/summer-strongest/fall) and time of highest emission (there likely is a shift in the exact time of year between highest emissions of pollen and spores and BC; because of different sampling resolution used, this might however not be possible to investigate in more detail), but this is already out of scope of the manuscript. Important is, that seasonality in their signal exists (and is preserved) allowing to count annual layers with both parameters yielding a comparable number of peaks (years), see related reply to referee Bohleber. For the reasons outlined above, a scatter plot would thus not be helpful or make much sense in the context of this study.

**Referee #4** Figure 3 shows a fair correspondence. Can the Authors plot a moving average line to better identify the peaks in 210Pb at Silvretta and Adamello?

**Authors:** This is not so easy because of the different sampling resolution of the two records. What should an objective averaging window be? How to treat the points of very high activity at the surfaces? etc…

Since we see no benefit from adding a trend line for the purpose of this figure, we prefer to keep it as simple as possible and thus in the current version. We are encouraged in this decision because based on the current visualization the referee agrees that "a fair correspondence" between the two records exists. This is the main and only take-home message. In the manuscript we accordingly write: "The ADA16 $^{210}$Pb record strongly resembles the $^{210}$Pb profile of the nearby Silvretta (SI) ice core…" and "…a reasonable alignment of the two $^{210}$Pb profiles was achieved, both showing a very similar, characteristic pattern…".

**Referee #4** Figure 5 is quite problematic. With just three points in the 1-40 years range it seems difficult to fit the Dansgaard Johnsen flow model up to 10000 years also considering the morphology of the bedrock underneath Pian di Neve. So I agree with the Author's comment at line 260-261. I would add 'very crude'.

**Authors:** We agree. Also considering the altitude of the site, we consider an age of up to 10000 to be very unlikely (see Bohleber et al. 2020). We removed the according numbers from the manuscript text. Considering also the comments from the other reviewers (see related comments there), the text of this paragraph has been reformulated in order to more clearly portray the main message. It now reads:

"For an estimation of the potential age range accessible by the Adamello ice archive, the one-dimensional Dansgaard-Johnsen ice-flow model was applied (Dansgaard and Johnsen, 1969). For the resulting age-depth relationship estimate shown in Figure 5 (Fig 6 in the revised version), model parameters were as follows. Based on the bedrock depth determined by ground penetrating radar measurements by Picotti et al. (2017), the value for glacier thickness at the drill site was $265 \pm 5$ m ($238 \pm 4.5$ m w.e.). The bottom shear zone thickness was assumed to be 15 % of the glacier thickness. This is slightly lower than the ~20 % typically observed for cold and polythermal high-elevation glaciers (e.g. Jenk et al., 2009; Uglietti et al., 2016; Gabrielli et al., 2016; Licciulli et al., 2020) but likely more reasonable for a temperate glacier (e.g. Kaspari et al., 2020). In any case, because constraining information from dated age horizons is lacking for the bottom part, a relatively large uncertainty of $\pm 10$ % was assigned. With these parameter settings, the value for the annual accumulation rate was found by tuning for a best model-fit to the dated 1986 and 1963 137Cs horizons (least squares approach). The dating uncertainty and the uncertainties of the pre-set model parameters as indicated above were employed to derive upper and lower bound estimates (to transfer the contribution form uncertainty in ice thickness to uncertainty in age, relative depths were used).

The model - nicely matching the determined bottom age for the ADA16 core and accounting for layer thinning (vertical strain) – provides us a best estimate of the mean annual accumulation rate at the ADA16 drill site for the period ~1946 to 1986 of $0.9 \pm 0.03$ m w.e. a-1. However, the assumption of steady-state conditions and the complexity of bedrock geometry and glacial flow in the deepest part of high-alpine glaciers strongly limits a realistic modelling of strain rates (and thus age) for the deeper parts, even with the most complex glaciological 3D ice-flow models. In our case, the lack of data for additional constraint in the deeper/older part, the assumption of steady-state conditions in annual accumulation rates (equal to an average value for the entire period contained in

the archive) which are further based on a relatively short time range covered by the 46 m core only, the derived model-based age-depth relationship can only yield a current best estimate. Anyhow, this is at least sufficient to reveal the potential of the site. Being contained in the major part of the total ice thickness (about the upper 240 m of ice; ~220 m w.e.), a millennial-long record should thus be accessible in high resolution. Also, there is reasonable likelihood for a few thousand more years contained in the remaining ~10 % of ice below. This is of high relevance in the perspective of an upcoming drilling campaign at Pian di Neve to retrieve an ice core down to bedrock."

**Referee #4** Conclusions. In the conclusions I would better stress the estimated accumulation rate of 0.8-0.9 m w.e. yr-1 which is quite convincing than the Dansgaard-Johnsen model age estimate which is very uncertain.

**Authors:** As suggested, we added the estimated accumulation also in the conclusion and at the same time weakened the statement regarding the age of the bottom ice, not giving a specific number.

**References not included in the manuscript**

Avak SE, Schwikowski M and Eichler A: Impact and implications of meltwater percolation on trace element records observed in a high-Alpine ice core. Journal of Glaciology 64(248), 877–886, doi: 10.1017/jog.2018.74, 2018.

Avak SE and 7 others: Melt-induced fractionation of major ions and trace elements in an Alpine snowpack. Journal of Geophysical Research: Earth Surface 124, 1647–1657, doi: 10.1029/2019JF005026, 2019.

Bohleber P, Schwikowski M, Stocker-Waldhuber M. et al.: New glacier evidence for ice-free summits during the life of the Tyrolean Iceman. Sci Rep 10, 20513 https://doi.org/10.1038/s41598-020-77518-9, 2020.

---

## Author Response (AR1)

Innsbruck, 18.06.2021

Dear Delphine Lannuzel, Editor of The Cryosphere

I am happy to submit, in behalf of all authors, the revised version of our manuscript entitled „*Significant mass loss in the accumulation area of the Adamello glacier indicated by the chronology of a 46 m ice core*".

The manuscript has been improved according to the comments provided by the reviewers and we hope this enhanced version will be well received.

Sincerely,
Daniela Festi

---

## Editor Decision (ED1)

[revised manuscript text omitted]

Figure 5Figure 6. Model based estimate of theled age-depth relationship down to bedrock for for the ADA16 drill site using the Dansgaard-Johnsen glacier flow model. Red dots show the 1986 and 1963 137Cs horizons used to fit the model. The estimated age at for the bottom of the 46 m long ADA16 ice core is shown shown in addition but was not used for the tuning (open diamond, not used for model tuning). The shaded area indicates the rangethe 95% confidence interval of estimates as confined by the upper and lower uncertainty bounds (thin blue lines).

**5    Conclusions**

Thanks to a combination of methods we succeeded in building a reliable timescale for the 46 m deep Adamello ice core ADA16, using ALC based on pollen and rBC, as well as the radioactive decay of $^{210}$Pb and time markers identified by maxima in $^{137}$Cs. According to the chronology we propose, the ADA16 ice core covers a period of about 50 years from ~around 1944 to 1995 1993 AD. , and that inFor the period 1963-86 anthe average annual net accumulation rate is of 0.859 ± 0.03 m w.e. ya$^{-1}$ could be determined. On the other hand, no accumulation was

preserved for about the last 20 years at the Adamello 2016 drilling site on Pian di Neve (Italian Alps), as indicated by all approaches used to estimate the age of the surface, yielding  an age older than the drilling date of 2016 This result is consistent with previous studies on mass balance in the region and suggests that the coring site is currently located below the  ELA. The lack of retained accumulation across the former  accumulation zone indicates a glacier that cannot survive (Pelto, 2010)clearly the Adamello glacier is clearly at high risk under present-day conditions . Our results undoubtedly highlight the fact that recent warming does not only compromise signal preservation, but can also  lead to  severe loss of glacier surface mass, even at high altitude. T removal of  annual accumulation layers  prevents the use of the drilling date as an anchor point for annual layer counting. Here, we demonstrated that establishing a chronology is nevertheless possible when using tracers in particulate form (pollen and spores, BC) or attached to particles ($^{210}$Pb), as they show to be least affected by melting. Finally, our results are encouraging leave room ofabout the possibilityt least somealso particularlythatthe Dansgaard-Johnsen glacierling approachwe appliedcould(6 kyr to >15 kyr)
[revised manuscript text omitted]

530

---

## Author Response (AR3)

Dear Delphine Lannuzel,

We are very happy to read that you find our manuscript "Significant mass loss in the accumulation area of the Adamello glacier indicated by the chronology of a 46 m ice core" suitable for publication in The Cryosphere.

We now made all the technical changes suggested in your review, and we here attach the track changes version of the manuscript.

Finally, we would like to thank you for your time and efforts in the review process.

Sincerely,

Daniela Festi, in behalf of all authors

[revised manuscript text omitted]